# Adaptive Position/Force Control of a Robotic Manipulator in Contact with a Flexible and Uncertain Environment

Piotr Gierlak 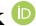

Department of Applied Mechanics and Robotics, Faculty of Mechanical Engineering and Aeronautics, Rzeszow University of Technology, al. Powstańców Warszawy 12, 35-959 Rzeszów, Poland; pgierlak@prz.edu.pl; Tel.: +48-17-865-1854

**Abstract:** The present paper concerns the synthesis of robot movement control systems in the cases of disturbances of natural position constraints, which are the result of surface susceptibility and inaccuracies in its description. The study contains the synthesis of control laws, in which the knowledge of parameters of the susceptible environment is not required, and which guarantee stability of the system in the case of an inaccurately described contact surface. The novelty of the presented solution is based on introducing an additional module to the control law in directions normal to the interaction surface, which allows for a fluent change of control strategy in the case of occurrence of distortions in the surface. An additional module in the control law is perceived as a virtual viscotic resistance force and resilient environment acting upon the robot. This interpretation facilitates intuitive selection of amplifications and allows for foreseeing the behavior of the system when disturbances occur. Introducing reactions of virtual constraints provides automatic adjustment of the robot interaction force with the susceptible environment, minimizing the impact of geometric inaccuracy of the environment.

**Keywords:** robotic manipulator; adaptive control; Lyapunov stability; position/force control; flexible environment; uncertain environment





## 1. Introduction

In recent years, in the field of robotization, rapid growth has been observed in applications of manipulator robots in tasks that require simultaneous execution of a desired trajectory and the interaction force of the end-effector with the environment. In terms of industrial applications, this refers to, inter alia, questions regarding the robotization of mechanical processing [1–4] or certain assembly tasks, but also a complex issue of the cooperation between robots and humans. A tendency towards precise force control, necessary for the correct execution of the assigned processes, is observed. In the field of industrial applications, these processes are mainly grinding [5], beveling [6], polishing [7,8], assembly [9,10], and friction welding [11,12].

The performance of the aforementioned tasks requires simultaneously executing movement in one direction and exerting force in the other direction. For instance, polishing of a surface may be performed as a result of the movement of the polishing tool on the surface and exerting force on it at the same time. From the perspective of the theory of robot control, the discussed problems refer to the control of systems with partial movement constraints [13,14]. From the point of view of mechanics, this is the solution to the problem of inverse dynamics of the system with imposed geometrical constraints. In the case of a manipulator robot moving in a free working space, that is, without contact between the end-effector and the environment, there are no constraints regarding the robot's position and it cannot develop any forces. If contact with the environment occurs, the movement of the robot is constrained and interaction forces, which need to be controlled, occur at the contact point.

Each task connected with the contact between a manipulator robot and the environment is characterized by a set of constraints, which result from the mechanical and geometric characteristics of the task [15]. For every task, the so-called generalized surface with position constraints along directions normal to this surface and force constraints along directions tangential to them may be defined. Such constraints divide possible movements of the end-effector into two orthogonal sets, whereas these movements must be controlled in accordance with different criteria. Friction forces acting in the tangential surface and resulting in the disturbance of natural force constraints constitute disturbances in the position control system and reduce the quality of control. In practice, natural position constraints may also not be rigidly maintained, e.g., due to susceptibility or inaccuracies in the description of the contact surface. Therefore, local movements of the end-effector in normal directions, which are the result of exerting force onto the surface, may occur. Significant and unknown susceptibility of the environment occurs in the case of, among others, the processing of thin-walled component parts. By comparison, the case of an inaccurately described surface of the environment originates from robotization of the processing of castings, which due to the phenomena connected with shrinking during solidification, have significant size and shape deviations.

Currently in industrial robotics, two main strategies for force control are applied in robotized processes of mechanical processing [16]. The first is based on maintaining the desired interaction force with a constant motion velocity and the trajectory is automatically adjusted to the shape of the contact surface (Figure 1a). A disadvantage of such a solution reveals itself in the example described below. If the processed surface is processed with the use of a tool with a small contact surface and if there is a cavity in the surface, it will be deepened with each passing of the tool, which is a consequence of exerting a constant pressure force and automatic adjustment of the trajectory to the surface [17]. Nonetheless, this method gives good results in applications such as polishing or grinding, when in the processed surface there are no significant cavities and the tool has a large operative surface [16].

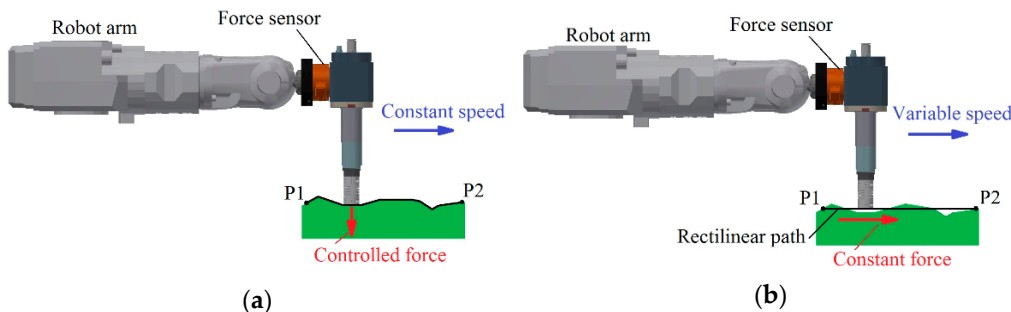

**Figure 1.** Main strategies for force control applied in industrial robotics: (**a**) The strategy of maintaining a given pressure force; (**b**) the strategy of adjusting the feed speed to the motion resistance.

The latter control strategy [16] is based on the movement of the end-effector of the robot along the desired trajectory regardless of the shape of the processed surface (Figure 1b). The variable value is velocity, which is dependent on the resistances of movement. If there are significant allowances in the processed surface, large resistances of movement occur and the velocity decreases. In this strategy, the pressure force is not a controlled value. It depends on the desired trajectory and the actual shape of the contact surface. This strategy is similar to the methods of processing with the use of a CNC [18], but the difference is that the advance velocity is dependent on tangential forces. A disadvantage of this method is that there is a possibility of the lack of contact between the end-effector and the surface if the desired trajectory diverges significantly from the actual location of the contact surface, which may lead to system failures [16]. Sometimes, basic control strategies in commercial applications are modified to introduce a certain flexibility in their execution [4]. Nonetheless, they are still two separate strategies.

Disadvantages of these solutions derive from the fact that in each of them only one criterion is taken into consideration, on the basis of which the control error is defined. This is the desired interaction force or the assigned shape of the interaction surface [16]. In the case of performing certain procedures of mechanical processing, mainly finishing processes, controlling the pressure force of the tool and taking into consideration the desired shape of the processed surface at the same time is vital [7]. Moreover, in the case of processing a surface of high susceptibility, it is necessary to take its deformation into consideration [19]. The currently developed and applied force control strategies fail to allow for taking these requirements into consideration. Furthermore, in commercial applications, it is necessary to position the processed surface against the robot with precision. In the case of failure to comply with this requirement, e.g., while processing component parts of significant inaccuracy in terms of their shapes and sizes, system failures occur [16]. One of the most typical system failures occurs when the end-effector of the robot has no contact with the processed surface in the precisely defined environment of the desired trajectory. The robot increases velocity and a sudden contact with a detail takes place or the driver cancels the task.

In the field of robotized mechanical processing [4,7,14], an important and topical issue is the development and implementation of a strategy for movement control, which could provide an appropriate quality of mechanical processing, despite the occurrence of phenomena that are not modelled, e.g., caused by significant errors in the description of the geometry of the processed component parts (connected with the uncertainty of their location regarding the robot) or local disturbances of their surface [20–29]. The present paper is concerned with the synthesis of movement control systems in the cases of disturbances of natural position constraints, which are the result of surface susceptibility and inaccuracies in its description. The study includes the synthesis of control laws, in which the knowledge of parameters of the susceptible environment is not required, and which guarantee stability of the system in the case of an inaccurately described surface.

## 2. New Approach to the Problem

In the paper, a new approach to the problem of control of a robot in interaction with the environment is described. A control strategy that combines two basic strategies is proposed. Nevertheless, this approach is not a simple combination of two strategies known from scientific writings [17,30] or industrial applications [16]. One of the component strategies is based on maintaining the desired interaction force with a constant motion velocity, while the trajectory is automatically adjusted to the shape of the contact surface. The second component strategy is based on executing the desired trajectory regardless of the shape of the processed surface. The combination of these two strategies on the basis of cooperation introduces cooperation between them, therefore, the requirements of each of the strategies are executed in a "soft" way. The proposed method may be closer to either of the strategies, depending on the introduced project factor. This is the amplification factor, which is responsible for regulating the priority to exert force or position.

If the theoretical description of the surface and the programmed trajectory in the normal direction that results from it are consistent with the actual shape of the processed surface, the first basic strategy is executed (Figure 2a). This is an extreme case of system operation and control is aimed only at the control of the down force in a direction normal to the surface.

If the shape of the surface diverges from theoretical assumptions, the importance of the second basic strategy, the activity of which is dependent on the differences between the programmed trajectory and the actual trajectory in the normal direction, that is, the trajectory execution error $\delta$. This is executed by introducing an additional control module $U_v$, which may be understood as a reaction of the virtual interaction surface. This complements the actual reaction force in the situation of an unforeseen change in the shape of the processed surface (Figure 2b). The control error is defined in a way that leads to the cooperative connection of both control strategies, which means that in the case of

distortions in the shape of the surface, none of the basic strategies will be fully executed, but the state of "balance" between the strategies will be achieved. This protects the system from the "extreme" activity of each of them. The disadvantages of basic strategies, which would occur if they were applied individually, are not revealed. This also allows using the advantages of the two basic strategies by means of fluent passage from one to the other. The activity of component control strategies and the result of their cooperative combination are presented in Figure 3.

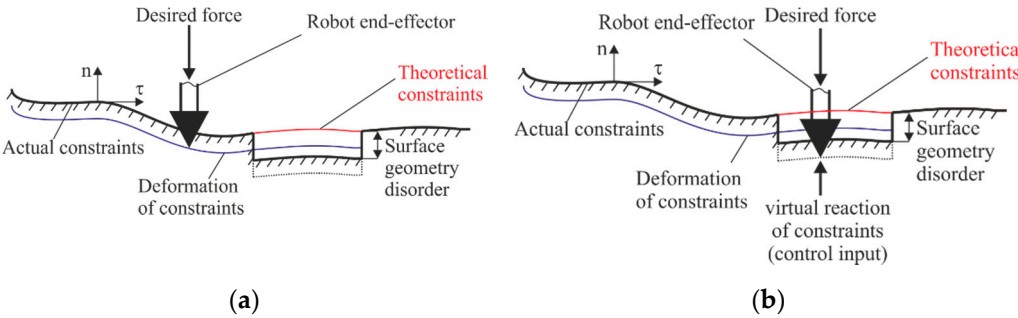

(**a**)                                                                                                 (**b**)

**Figure 2.** The activity of control strategy in the case of distortions in the surface: (**a**) Movement on the surface without distortion; (**b**) movement on the distorted surface.

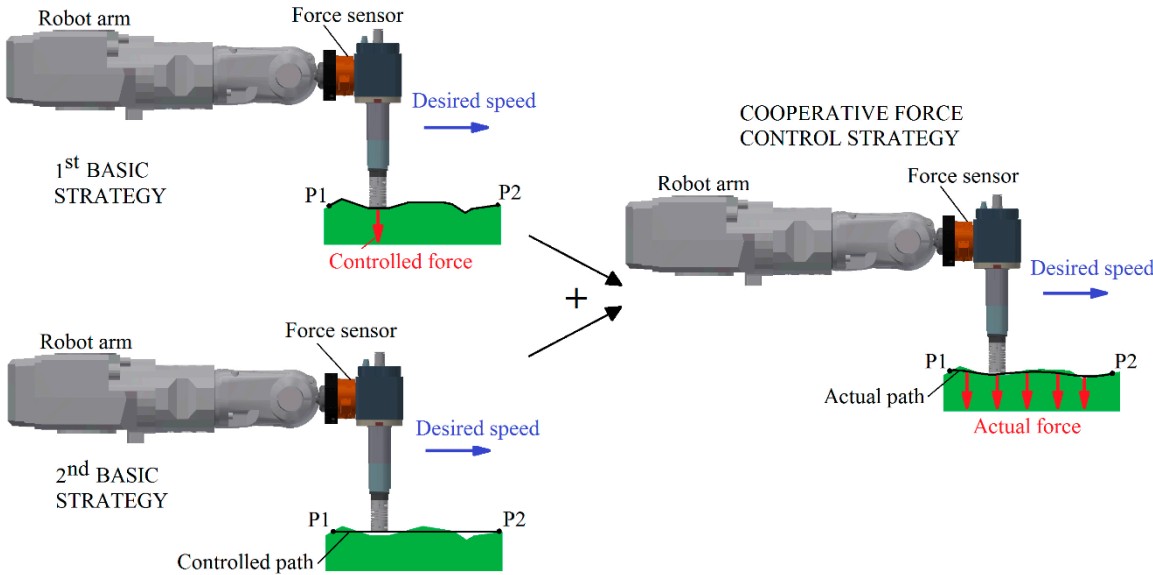

**Figure 3.** Cooperative strategy for force control.

The cooperative control strategy is not a combination of the two discussed strategies used in commercial industrial applications [16], but is a new approach to the problem. The first basic strategy corresponds with the strategy used in commercial industrial applications (Figure 1a); it also has solid theoretical foundations and is well established in the literature [16,17,22–25,30]. The second basic strategy fails to correspond with the presented commercial solution (Figure 1b). It originates from the position control methods [17] and its aim is to execute a trajectory that would provide an appropriate shape of the processed surface.

## 3. Dynamics of the Robot—Flexible Environment System

From the perspective of the task being performed, which is defined in the task space, it is convenient to present the dynamics of the robotic manipulator in the same space. In most

practical applications, the task space is related to Cartesian space. Thus, in the discussion below, the description of the robot's dynamics in task space is used. The dynamic equations of the motion of a robotic manipulator in this space can be represented in the form [19]:

$$A(q)E\ddot{\theta} + H(q,\dot{q})E\dot{\theta} + B(q,\dot{q}) + \Psi(q,t) = U + \Lambda, \tag{1}$$

where $q \in R^n$—vector of generalized coordinates (joint coordinates), $\theta \in R^m$—vector of task variables, $A(q) \in R^{m \times m}$—inertia matrix, $H(q,\dot{q}) \in R^{m \times m}$ matrix of coefficients of centrifugal and Coriolis forces, $B(q,\dot{q}) \in R^m$—vector of friction and gravitational forces, $\Psi(q,t) \in R^m$—vector of limited interference, $U \in R^m$—vector of control inputs, $\Lambda \in R^m$—vector of interaction forces of the robot with the environment, $E \in R^{m \times m}$—matrix of system vulnerability, $m$—the dimension of the task space (it was assumed that the dimension of the task space is equal to the dimension of the robot joint).

The vector of interaction forces has the following form:

$$\Lambda = \begin{bmatrix} F_{e\tau} \\ F_{en} \end{bmatrix}, \tag{2}$$

where $F_{en} \in R^r$—vector of normal forces, $F_{e\tau} \in R^{m-r}$—vector of tangent forces. The vector of task variables has the following form:

$$\theta = \begin{bmatrix} c_\tau \\ F_{en} \end{bmatrix} \in R^m, \tag{3}$$

where $c_\tau \in R^{m-r}$—vector of tangential displacement. This decomposition of the vector of interaction forces $\Lambda$ and the vector of task variables $\theta$ results from the decomposition of the $m$-dimensional task space $\{C\}$ into $r$-dimensional normal subspace $\{N\}$ and $(m-r)$-dimensional tangential subspace $\{T\}$ [31,32]. The vulnerability of the surfaces limiting the movement of the robot tip is included in the dynamics description because the system vulnerability matrix has the following form:

$$E = \begin{bmatrix} I_{(m-r) \times (m-r)} & 0 \\ 0 & P_e \end{bmatrix} \in R^{m \times m}, \tag{4}$$

where $P_e = R^{r \times r}$—the environmental vulnerability matrix.

Equation (1) describes the system dynamics in task coordinates and is a function of kinematic motion parameters in tangential directions and forces in normal directions. The presented description of the dynamics of the system in the task space related to the surface of the environment facilitates the definition and implementation of the task to be performed by the robotic manipulator, i.e., the implementation of the motion with the pressure of the end-effector. The simultaneous introduction of variables related to motion and normal forces of interaction to the task variables vector describing the system state allows the use of the same control methods, e.g., with regard to non-linearity compensation for both position and force control. The paper [19] provides information on the transformation of the dynamics description from the robot's configuration space to the task space, and then the decomposition of the task space into tangential and normal subspaces.

## 4. Adaptive Position/Force Tracking Control

This section proposes a control strategy that can be called cooperative because it combines two different control strategies on the basis of cooperation (Figure 3). The purpose of this approach is to supplement one strategy by the other in situations in which a given elementary strategy applied individually leads to unfavorable behavior of the robot. To properly implement the strategy, the work involved the direct measurement of interaction forces using a force sensor placed in the robot's end-effector. This ensures correct implementation of the feedback loop in the force control, even when the surface of the

environment is not known exactly. The adaptive control algorithm presented in this section is based on the assumption of knowledge of the structure of the system dynamics model. Implementation of a cooperative control strategy requires the assumption of nominal contact surface geometry (determination of theoretical constraints), desired trajectory of motion and force, and knowledge of the current position of the robot's end-effector.

**Assumption 1.** *For dynamical system (1), the following trajectories are given:*

- *Limited trajectory of motion of the robot's end-effector in the tangential plane $c_{\tau d}(t) \in R^{m-r}$, $\dot{c}_{\tau d}(t), \ddot{c}_{\tau d}(t)$;*
- *Limited force trajectory in normal directions $F_{end}(t) \in R^r, \dot{F}_{end}(t), \ddot{F}_{end}(t)$;*
- *Limited nominal trajectory of motion of the robot's end-effector in normal directions $c_{n\ nom}(t) \in R^r, \dot{c}_{n\ nom}(t), \ddot{c}_{n\ nom}(t)$, which results from the assumed surface shape.*

Assumption 1 concerning simultaneous knowledge of the description of the nominal motion path and the force trajectory in the same direction (normal) is a significant difference in comparison to the assumptions formulated in typical issues related to position/force control. It also allows the definition of a modified control objective, by an appropriate definition of the filtered tracking error, which in the case of taking into account the inaccuracy of constraints must be different than in the case of knowing the environment surface or omitting its inaccuracy.

To define the control objective, control errors were introduced, where:

$$\widetilde{c}_\tau = c_{\tau d} - c_\tau, \tag{5}$$

is the error of motion in the tangential plane, and:

$$\widetilde{F}_{en} = F_{end} - F_{en}, \tag{6}$$

is the error of force in the normal direction. An auxiliary variable $\xi \in R^r$ is defined such that:

$$\delta = c_n - c_{n\ nom} - \delta_0, \tag{7}$$

which is related to the difference between the nominal position of the robot end-effector $c_{n\ nom}$ resulting from the theoretically existing constraints and the real position $c_n$ in the normal direction. That is, $\delta$ is the deviation of the end-effector from the assumed constraints in the normal direction. In detail, it should be added that the expression $\delta_0 = K_e^{-1} F_{end}$ is the predicted surface deformation derived from the pressure force $F_{en}$. With high surface rigidity, this deformation is negligible, but for more flexible elements its inclusion is necessary.

In the case of disturbances of geometrical constraints, the system will be stimulated by disturbances related to surface inaccuracies. To illustrate this, Figure 4 shows the case when the end-effector of the robotic manipulator remains in contact with the surface of the environment, but the shape of this surface does not correspond to the nominal path $c_{n\ nom}$.

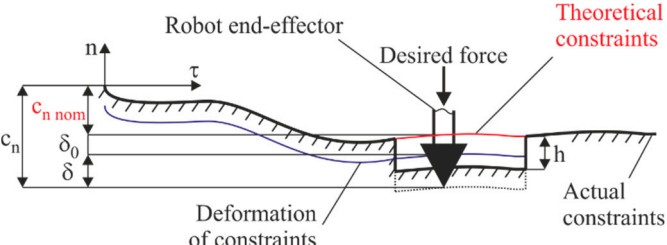

**Figure 4.** The operation of the cooperative force control strategy in the case of surface inaccuracy ($h$ is a variable characterizing the inaccuracy of the real contact surface).

The aim of the control is to follow the trajectory of motion, i.e., to minimize the motion error $\widetilde{c}_\tau$, and to follow the force trajectory, i.e., to minimize the force error $\widetilde{F}_{en}$ if the real and nominal constraints match ($\delta = \mathbf{0}$). In the event of their non-compliance, the goal is to minimize the error $\widetilde{F}_{en} - w_\delta \delta$, which is a combination of force error and deviation error from the nominal constraints. This allows for the introduction of a cooperative control strategy in the normal direction, which for the co-operation gain matrix $w_\delta = \mathbf{0}$ changes into the force control strategy, whereas for $w_\delta > \mathbf{0}$ the importance of the strategy for maintaining the nominal motion path in the normal direction is strengthened in the case of imperfections in surface contact. If the real and nominal constraints match, the deviation $\delta = \mathbf{0}$ and only the force control strategy is implemented independently of the $w_\delta$ matrix.

To achieve the control objective, a filtered tracking error was introduced:

$$s = \begin{bmatrix} s_\tau \\ s_n \end{bmatrix}, \tag{8}$$

in which:

$$s_\tau = \dot{\widetilde{c}}_\tau + \Lambda_\tau \widetilde{c}_\tau, \tag{9}$$

$$s_n = \dot{\widetilde{F}}_{en} - w_\delta \dot{\delta} + \Lambda_n \left( \widetilde{F}_{en} - w_\delta \delta \right), \tag{10}$$

where $\Lambda_\tau \in R^{(m-r) \times (m-r)}$ and $\Lambda_n \in R^{r \times r}$ are diagonal gain matrices, $w_\delta \in R^{r \times r}$ is the co-operation gain matrix. Expression (10) introduces a new approach to the problem of control in the normal direction to the surface of constraints. Compared to the approach presented in papers [19,33], in which the generalized error $s_n$ for normal directions depends on the force error and its derivative, here it also depends on deviation $\delta$ from the desired nominal motion path in the normal direction and from the derivative $\dot{\delta}$.

Equations (8)–(10) make it possible to write the filtered tracking error with the form:

$$s = \begin{bmatrix} \dot{\widetilde{c}}_\tau \\ \dot{\widetilde{F}}_{en} - w_\delta \dot{\delta} \end{bmatrix} + \begin{bmatrix} \Lambda_\tau & \mathbf{0} \\ \mathbf{0} & \Lambda_n \end{bmatrix} \begin{bmatrix} \widetilde{c}_\tau \\ \widetilde{F}_{en} - w_\delta \delta \end{bmatrix}, \tag{11}$$

and then convert it to the form:

$$\dot{\theta} = -s + \begin{bmatrix} \dot{c}_{\tau d} \\ \dot{F}_{end} - w_\delta \dot{\delta} \end{bmatrix} + \begin{bmatrix} \Lambda_\tau & \mathbf{0} \\ \mathbf{0} & \Lambda_n \end{bmatrix} \begin{bmatrix} \widetilde{c}_\tau \\ \widetilde{F}_{en} - w_\delta \delta \end{bmatrix}, \tag{12}$$

By introducing Equation (12) into Formula (1), a description of the dynamics as a function of the filtered tracking error was obtained:

$$A(q)E\dot{s} = -H(q, \dot{q})Es + A(q)E\dot{v} + H(q, \dot{q})Ev + B(q, \dot{q}) + \Psi(q, t) - U - \Lambda, \tag{13}$$

where there is an auxiliary variable that has the form:

$$v = \begin{bmatrix} \dot{c}_{\tau d} \\ \dot{F}_{end} - w_\delta \dot{\delta} \end{bmatrix} - \begin{bmatrix} \Lambda_\tau & \mathbf{0} \\ \mathbf{0} & \Lambda_n \end{bmatrix} \begin{bmatrix} \widetilde{c}_\tau \\ \widetilde{F}_{en} - w_\delta \delta \end{bmatrix}. \tag{14}$$

The non-linear part of Equation (13) is designated as:

$$A(q)E\dot{v} + H(q, \dot{q})Ev + B(q, \dot{q}) = f, \tag{15}$$

where $f \in R^m$ is a function dependent on both the robot mathematical model and the environment. In particular, it depends on unknown robot parameters and unknown rigidity of the environment, but also on its possible inaccuracies. Finally, the description of the dynamics of the system has the following form:

$$A(q)E\dot{s} = -H(q, \dot{q})Es + F + \Psi(q, t) - U - \Lambda. \tag{16}$$

Next, the control law including the PD controller, the non-linearity compensating control $\hat{f} \in R^m$, the term compensating the interaction force $\Lambda$, and the robust term $r \in R^m$ was assumed:

$$U = K_D s + \hat{f} - \Lambda - r, \tag{17}$$

In the control law, the expression $K_D s$ is a form of PD control, where $K_D \in R^{m \times m}$ is a gain matrix such that $K_D = K_D^T > 0$, and function $\hat{f}$ approximates $f$. The third part of the control law introduces feedback from the forces of interaction of the robotic manipulator with its environment in all directions. It is important to clarify the form of the first two parts of the control law that introduce differences in relation to the typical control laws. Regarding the first control part, it is possible to decompose the $K_D$ matrix according to the equation:

$$K_D = \begin{bmatrix} K_{D\tau} & 0 \\ 0 & K_{Dn} \end{bmatrix}, \tag{18}$$

and the $K_D s$ part is written taking into account using Equation (11) as:

$$K_D s = \begin{bmatrix} K_{D\tau} & 0 \\ 0 & K_{Dn} \end{bmatrix} \begin{bmatrix} \dot{\tilde{c}}_\tau \\ \dot{\tilde{F}}_{en} - w_\delta \dot{\delta} \end{bmatrix} + \begin{bmatrix} K_{D\tau} & 0 \\ 0 & K_{Dn} \end{bmatrix} \begin{bmatrix} \Lambda_\tau & 0 \\ 0 & \Lambda_n \end{bmatrix} \begin{bmatrix} \tilde{c}_\tau \\ \tilde{F}_{en} - w_\delta \delta \end{bmatrix}, \tag{19}$$

where $K_{D\tau} \in R^{(m-r) \times (m-r)}$ and $K_{Dn} \in R^{r \times r}$ are diagonal gain matrices. Then, Equation (19) is written in the form:

$$K_D s = \begin{bmatrix} K_{D\tau} \dot{\tilde{c}}_\tau + K_{D\tau} \Lambda_\tau \tilde{c}_\tau \\ K_{Dn} \dot{\tilde{F}}_{en} + K_{Dn} \Lambda_n \tilde{F}_{en} - K_{Dn} w_\delta \dot{\delta} - K_{Dn} \Lambda_n w_\delta \delta \end{bmatrix}, \tag{20}$$

from which it follows that, in comparison to a conventional force control containing only the elements $K_{Dn} \dot{\tilde{F}}_{en} + K_{Dn} \Lambda_n \tilde{F}_{en}$, the equation also has the part—$K_{Dn} w_\delta \dot{\delta} - K_{Dn} \Lambda_n w_\delta \delta$. It can be written in the form $-K_c \dot{\delta} - K_k \delta$. Then, $K_k = K_{Dn} \Lambda_n w_\delta$ is the effective proportional gain coefficient, which can be understood as the coefficient of the virtual resilient resistance force, and $K_c = K_{Dn} w_\delta$ is the effective differential gain coefficient, which can be understood as the coefficient of the virtual viscous damping force. To summarize, it should be stated that, in the case of surface inaccuracy, the control $-K_c \dot{\delta} - K_k \delta$ becomes active, replacing the surface effect in such a way as if it had a coefficient of resilience $K_k$ and damping $K_c$.

The part $\hat{f}$ was introduced into the control law of Equation (17) to compensate for the non-linear function $f$, which depends on, inter alia, the inaccuracy of the surface $\delta$, not just from the force error $\tilde{F}_{en}$. The non-linear function can be decomposed into two parts, one of which corresponds to the tangential directions and the other the normal direction:

$$f = \begin{bmatrix} f_\tau \\ f_n \end{bmatrix}, \tag{21}$$

where:

$$\left. \begin{array}{c} f_\tau = D_\tau f \in R^{(m-r)} \\ f_n = D_n f \in R^r \\ D_\tau = \begin{bmatrix} I_{(m-r) \times (m-r)} & 0 \end{bmatrix} \in R^{(m-r) \times m} \\ D_n = \begin{bmatrix} 0 & I_{r \times r} \end{bmatrix} \in R^{r \times m} \end{array} \right\}, \tag{22}$$

The constituent functions $f_\tau$ and $f_n$ described by System (22) can be approximated with the help of various techniques [34–38]. In the next subsection, one of these, which is an adaptation method, is presented.

The remainder of this section describes the stability of the adaptive control algorithm, which was introduced to avoid the problem of not knowing the object parameters, assuming knowledge of the structure of mathematical model of object.

Using the linearity of the $f_\tau$ and $f_n$ functions with respect to the parameters, the following were written:

$$f_\tau = D_\tau A(q) E\ddot{v} + D_\tau H(q, \dot{q}) Ev + D_\tau B(q, \dot{q}) = Y_\tau(q, \dot{q}, v, \ddot{v}) p_\tau, \tag{23}$$

$$f_n = D_n A(q) E\ddot{v} + D_n H(q, \dot{q}) Ev + D_n B(q, \dot{q}) = \begin{bmatrix} Y_{n1}(q, \dot{q}, v, \ddot{v}) p_{n1} \\ \vdots \\ Y_{ni}(q, \dot{q}, v, \ddot{v}) p_{ni} \\ \vdots \\ Y_{nr}(q, \dot{q}, v, \ddot{v}) p_{nr} \end{bmatrix}, \tag{24}$$

where the matrices $Y_\tau(q, \dot{q}, v, \ddot{v}) \in R^{(m-r) \times B\tau}$ and $Y_{ni}(q, \dot{q}, v, \ddot{v}) \in R^{1 \times bni}$ are called regression matrices, $p_\tau \in R^{b\tau}$ and $p_{ni} \in R^{bni}$ are vectors of unknown parameters, $b\tau$, $bni$ are the dimensions of the parameter space, and $i = 1, \ldots, r$. The approximation of the functions $f_\tau$ and $f_n$ by $\hat{f}_\tau$ and $\hat{f}_n$ depends on the estimation of the vectors of unknown parameters, which is written as follows:

$$\hat{f} = \begin{bmatrix} \hat{f}_\tau \\ \hat{f}_n \end{bmatrix} = \begin{bmatrix} Y_\tau(q, \dot{q}, v, \ddot{v}) \hat{p}_\tau \\ Y_{n1}(q, \dot{q}, v, \ddot{v}) \hat{p}_{n1} \\ \vdots \\ Y_{ni}(q, \dot{q}, v, \ddot{v}) \hat{p}_{ni} \\ \vdots \\ Y_{nr}(q, \dot{q}, v, \ddot{v}) \hat{p}_{nr} \end{bmatrix}, \tag{25}$$

where:

$$\hat{f}_\tau = Y_\tau(q, \dot{q}, v, \ddot{v}) \hat{p}_\tau, \tag{26}$$

$$\hat{f}_{ni} = Y_{ni}(q, \dot{q}, v, \ddot{v}) \hat{p}_{ni}, \tag{27}$$

and $\hat{p}_\tau \in R^{b\tau}$, $\hat{p}_{ni} \in R^{bni}$ are parameter vector estimates.

Assuming the control law of Equation (17) and taking into account relationships of Equations (21)–(27), a description of a closed system was obtained in the form:

$$A(q) E\dot{s} = -H(q, \dot{q}) Es - K_D s + r + \begin{bmatrix} Y_\tau(q, \dot{q}, v, \ddot{v}) \widetilde{p}_\tau \\ Y_{n1}(q, \dot{q}, v, \ddot{v}) \widetilde{p}_{n1} \\ \vdots \\ Y_{ni}(q, \dot{q}, v, \ddot{v}) \widetilde{p}_{ni} \\ \vdots \\ Y_{nr}(q, \dot{q}, v, \ddot{v}) \widetilde{p}_{nr} \end{bmatrix} + \Psi(q, t), \tag{28}$$

where:

$$\widetilde{p}_\tau = p_\tau - \hat{p}_\tau, \tag{29}$$

$$\widetilde{p}_{ni} = p_{ni} - \hat{p}_{ni}, \tag{30}$$

are errors of parameter estimates. Formula (28) is a description of a closed control system as a function of the filtered tracking error and errors of parameter estimates.

In the case of inaccurate surfaces, the analysis of asymptotic stability has no practical justification, which is why proof of the practical stability of the control system is presented. To this end, appropriate assumptions were made.

**Assumption 2.** *There is limited interference on the dynamic System (1):*

$$\Psi(q, t) = \begin{bmatrix} \Psi_\tau(q, t) \\ \Psi_n(q, t) \end{bmatrix}, \tag{31}$$

*where* $\Psi_\tau(q,t) \in R^{m-r}$, $\Psi_n(q,t) \in R^r$, *and* $b_\tau$, $b_{ni}$ *are known constants such that* $||\Psi_\tau(q,t)|| \le b_\tau$ *and* $||\Psi_n(q,t)|| \le b_{ni}$.

**Assumption 3.** *The vector of filtered tracking error in the form of Equation (11) can be decomposed according to the equation:*

$$s = \begin{bmatrix} s_\tau \\ s_n \end{bmatrix}, \tag{32}$$

*where* $s_\tau \in R^{m-r}$, $s_n = \begin{bmatrix} s_{n1} & \dots & s_{ni} & \dots & s_{nr} \end{bmatrix}^T \in R^r$.

**Assumption 4.** *A robust term in the control law of Equation (17) can be decomposed in the following way:*

$$r = \begin{bmatrix} r_\tau \\ r_n \end{bmatrix}, \tag{33}$$

*where:*

$$r_\tau = -\frac{K_\tau}{s_\tau}s_\tau, \tag{34}$$

$$r_{ni} = -K_{ni}\frac{s_{ni}}{|s_{ni}|}, \tag{35}$$

*and* $K_\tau > b_\tau \ge ||\Psi_\tau(q,t)||$, $K_{ni} > b_{ni} \ge |\Psi_{ni}(q,t)|$.

**Assumption 5.** *The parameter vectors are limited so that:*

$$||p_\tau|| \le p_{\tau\,max}, \tag{36}$$

$$||p_{ni}|| \le p_{ni\,max}. \tag{37}$$

**Assumption 6.** *The parameter adaptation law takes the form of the equations [37,38]:*

$$\dot{p}_\tau = \Gamma_\tau Y_\tau\big(q,\dot{q},v,\dot{v}\big)^T s_\tau - k_\tau||s_\tau||\Gamma_\tau\hat{p}_\tau, \tag{38}$$

$$\dot{p}_{ni} = \Gamma_{ni} Y_{ni}\big(q,\dot{q},v,\dot{v}\big)^T s_{ni} - k_{ni}|s_{ni}|\Gamma_{ni}\hat{p}_{ni}, \tag{39}$$

*where* $\Gamma_\tau = \Gamma_\tau^T > 0$, $\Gamma_{ni} = \Gamma_{ni}^T > 0$ *are design adaptation gain matrices,* $k_\tau > 0$ *and* $k_{ni} > 0$ *are design parameters.*

**Theorem 1.** *If the dynamic system described by Equation (1) is controlled by Equation (17) and Assumptions 1–6 are fulfilled, the filtered tracking errors* $s_\tau$ *and* $s_{ni}$ *and estimation errors* $\tilde{p}_\tau$ *and* $\tilde{p}_{ni}$ *are uniformly ultimately bounded with practical limits given by the right-hand side of the Equations (55)–(58), respectively.*

**Proof.** The description of the system given by Equation (1) was transformed into a description in terms of the filtered tracking error of Equation (16), and after the introduction of the control law of Equation (17), a closed system description (Equation (28)) was obtained, including disturbances and a robust term. To demonstrate the stability of the closed system, the Lyapunov stability theory was used. The function was assumed:

$$V = \frac{1}{2}s^T E^T A(q)Es + \frac{1}{2}\tilde{p}_\tau^T \Gamma_\tau^{-1}\tilde{p}_\tau + \frac{1}{2}\sum_{i=1}^r P_{eii}\tilde{p}_{ni}^T \Gamma_{ni}^{-1}\tilde{p}_{ni}, \tag{40}$$

where $P_{eii}$ is an element of the matrix of environmental vulnerability. By calculating the derivative of Function (40) with respect to time the following was obtained:

$$\dot{V} = \frac{1}{2}s^T E^T \dot{A}(q)Es + s^T E^T A(q)E\dot{s} + \tilde{p}_\tau^T \Gamma_\tau^{-1}\dot{\tilde{p}}_\tau + \sum_{i=1}^r P_{eii}\tilde{p}_{ni}^T \Gamma_{ni}^{-1}\dot{\tilde{p}}_{ni}. \tag{41}$$

Taking into account Equation (28) in Equation (41) the following was written:

$$\dot{V} = \tfrac{1}{2}s^T E^T \left[ \dot{A}(q) - 2H(q,\dot{q}) \right] Es - s^T E^T K_D s + s^T E^T r + s^T E^T \Psi(q,t) + s^T E^T \begin{bmatrix} Y_\tau(q,\dot{q},v,\dot{v})\widetilde{p}_\tau \\ Y_{n1}(q,\dot{q},v,\dot{v})\widetilde{p}_{n1} \\ \vdots \\ Y_{ni}(q,\dot{q},v,\dot{v})\widetilde{p}_{ni} \\ \vdots \\ Y_{nr}(q,\dot{q},v,\dot{v})\widetilde{p}_{nr} \end{bmatrix} + \tag{42}$$

$$\widetilde{p}_\tau^T \Gamma_\tau^{-1}\dot{\widetilde{p}}_\tau + \sum_{i=1}^r P_{eii}\widetilde{p}_{ni}^T \Gamma_{ni}^{-1}\dot{\widetilde{p}}_{ni}.$$

Given that $E^T = E$, the fifth element of Equation (42) was transformed into the following form:

$$s^T E^T \begin{bmatrix} Y_\tau(q,\dot{q},v,\dot{v})\widetilde{p}_\tau \\ Y_{n1}(q,\dot{q},v,\dot{v})\widetilde{p}_{n1} \\ \vdots \\ Y_{ni}(q,\dot{q},v,\dot{v})\widetilde{p}_{ni} \\ \vdots \\ Y_{nr}(q,\dot{q},v,\dot{v})\widetilde{p}_{nr} \end{bmatrix} = s^T \begin{bmatrix} Y_\tau(q,\dot{q},v,\dot{v})\widetilde{p}_\tau \\ P_{e11}Y_{n1}(q,\dot{q},v,\dot{v})\widetilde{p}_{n1} \\ \vdots \\ P_{eii}Y_{ni}(q,\dot{q},v,\dot{v})\widetilde{p}_{ni} \\ \vdots \\ P_{err}Y_{nr}(q,\dot{q},v,\dot{v})\widetilde{p}_{nr} \end{bmatrix} \tag{43}$$

In addition, the expression $E^T \left[ \dot{A}(q) - 2H(q,\dot{q}) \right] E$ is a skew-symmetrical matrix (see Property 5 in [19]), therefore the following relationship emerges:

$$s^T E^T \left[ \dot{A}(q) - 2H(q,\dot{q}) \right] Es = 0. \tag{44}$$

Taking into account Equations (4), (32), (43) and (44), the following equation was obtained:

$$\dot{V} = -\begin{bmatrix} s_\tau^T & s_n^T \end{bmatrix} \begin{bmatrix} I_{(m-r)\times(m-r)} & 0 \\ 0 & P_e \end{bmatrix} K_D \begin{bmatrix} s_\tau \\ s_n \end{bmatrix} + \begin{bmatrix} s_\tau^T & s_n^T \end{bmatrix} \begin{bmatrix} I_{(m-r)\times(m-r)} & 0 \\ 0 & P_e \end{bmatrix} [\Psi(q,t) + r] +$$

$$\begin{bmatrix} s_\tau^T & s_n^T \end{bmatrix} \begin{bmatrix} Y_\tau(q,\dot{q},v,\dot{v})\widetilde{p}_\tau \\ P_{e11}Y_{n1}(q,\dot{q},v,\dot{v})\widetilde{p}_{n1} \\ \vdots \\ P_{eii}Y_{ni}(q,\dot{q},v,\dot{v})\widetilde{p}_{ni} \\ \vdots \\ P_{err}Y_{nr}(q,\dot{q},v,\dot{v})\widetilde{p}_{nr} \end{bmatrix} + \widetilde{p}_\tau^T \Gamma_\tau^{-1}\dot{\widetilde{p}}_\tau + \sum_{i=1}^r P_{eii}\widetilde{p}_{ni}^T \Gamma_{ni}^{-1}\dot{\widetilde{p}}_{ni} \tag{45}$$

and substituting Equations (31) and (33) the description was obtained:

$$\dot{V} = -\begin{bmatrix} s_\tau^T & s_n^T \end{bmatrix} \begin{bmatrix} I_{(m-r)\times(m-r)} & 0 \\ 0 & P_e \end{bmatrix} \begin{bmatrix} K_{D\tau} & 0 \\ 0 & K_{Dn} \end{bmatrix} \begin{bmatrix} s_\tau \\ s_n \end{bmatrix} + \begin{bmatrix} s_\tau^T & s_n^T \end{bmatrix} \begin{bmatrix} I_{(m-r)\times(m-r)} & 0 \\ 0 & P_e \end{bmatrix} \begin{bmatrix} \Psi_\tau(q,t) + r_\tau \\ \Psi_n(q,t) + r_n \end{bmatrix} +$$

$$\widetilde{p}_\tau^T \left[ Y_\tau(q,\dot{q},v,\dot{v})^T s_\tau + \Gamma_\tau^{-1}\dot{\widetilde{p}}_\tau \right] + \sum_{i=1}^r \widetilde{p}_{ni}^T \left[ s_{ni}P_{eii}Y_{ni}(q,\dot{q},v,\dot{v})^T + P_{eii}\Gamma_{ni}^{-1}\dot{\widetilde{p}}_{ni} \right], \tag{46}$$

in which the matrix $K_D$ was replaced by submatrices $K_{D\tau} \in R^{(m-r)\times(m-r)}$ and $K_{Dn} \in R^{r\times r}$ such that:

$$K_D = \begin{bmatrix} K_{D\tau} & 0 \\ 0 & K_{Dn} \end{bmatrix}. \tag{47}$$

Substituting the parameters adaptation laws of Equations (38) and (39), the following equation was obtained:

$$\dot{V} = -s_\tau^T K_{D\tau} s_\tau - s_n^T P_e K_{Dn} s_n + s_\tau^T \Psi_\tau(q,t) + s_\tau^T r_\tau + \sum_{i=1}^r s_{ni}P_{eii}\Psi_{ni}(q,t) + \sum_{i=1}^r s_{ni}P_{eii}r_{ni} + k_\tau ||s_\tau|| \widetilde{p}_\tau^T \hat{p}_\tau +$$

$$\sum_{i=1}^r P_{eii}k_{ni}|s_{ni}|\widetilde{p}_{ni}^T \hat{p}_{ni}. \tag{48}$$

Taking into account the following limitations:

$$\widetilde{\boldsymbol{p}}_\tau^T \hat{\boldsymbol{p}}_\tau = \widetilde{\boldsymbol{p}}_\tau^T (\boldsymbol{p}_\tau - \widetilde{\boldsymbol{p}}_\tau) = \widetilde{\boldsymbol{p}}_\tau^T \boldsymbol{p}_\tau - \widetilde{\boldsymbol{p}}_\tau^T \widetilde{\boldsymbol{p}}_\tau \leq ||\widetilde{\boldsymbol{p}}_\tau|| \, ||\boldsymbol{p}_\tau|| - ||\widetilde{\boldsymbol{p}}_\tau||^2 \leq ||\widetilde{\boldsymbol{p}}_\tau|| p_{\tau max} - ||\widetilde{\boldsymbol{p}}_\tau||^2, \tag{49}$$

$$\widetilde{\boldsymbol{p}}_{ni}^T \hat{\boldsymbol{p}}_{ni} = \widetilde{\boldsymbol{p}}_{ni}^T (\boldsymbol{p}_{ni} - \widetilde{\boldsymbol{p}}_{ni}) = \widetilde{\boldsymbol{p}}_{ni}^T \boldsymbol{p}_{ni} - \widetilde{\boldsymbol{p}}_{ni}^T \widetilde{\boldsymbol{p}}_{ni} \leq ||\widetilde{\boldsymbol{p}}_{ni}|| \, ||\boldsymbol{p}_{ni}|| - ||\widetilde{\boldsymbol{p}}_{ni}||^2 \leq ||\widetilde{\boldsymbol{p}}_{ni}|| p_{nimax} - ||\widetilde{\boldsymbol{p}}_{ni}||^2, \tag{50}$$

and robust term elements of Equations (34) and (35), Equation (48) was transformed into the form:

$$\dot{V} \leq -K_{D\tau min}||s_\tau||^2 - \sum_{i=1}^r P_{eii} K_{Dnii} |s_{ni}|^2 + k_\tau ||s_\tau|| \left( ||\widetilde{\boldsymbol{p}}_\tau|| p_{\tau\,max} - ||\widetilde{\boldsymbol{p}}_\tau||^2 \right) + \sum_{i=1}^r P_{eii} k_{ni} |s_{ni}| \left( ||\widetilde{\boldsymbol{p}}_{ni}|| p_{ni\,max} - ||\widetilde{\boldsymbol{p}}_{ni}||^2 \right), \tag{51}$$

where $K_{D\tau min}$ is the minimum singular value of $\boldsymbol{K}_{D\tau}$. Inequality (51) was transformed into the following form:

$$\dot{V} \leq -||s_\tau|| [K_{D\tau min}||s_\tau|| + k_\tau ||\widetilde{\boldsymbol{p}}_\tau|| (||\widetilde{\boldsymbol{p}}_\tau|| - p_{\tau\,max})] - \sum_{i=1}^r |s_{ni}| [P_{eii} K_{Dnii} |s_{ni}| + P_{eii} k_{ni} ||\widetilde{\boldsymbol{p}}_{ni}|| (||\widetilde{\boldsymbol{p}}_{ni}|| - p_{ni\,max})]. \tag{52}$$

The function $\dot{V}$ is negative if the expressions in square brackets are positive. These expressions are written as follows:

$$K_{D\tau min}||s_\tau|| + k_\tau ||\widetilde{\boldsymbol{p}}_\tau|| (||\widetilde{\boldsymbol{p}}_\tau|| - p_{\tau\,max}) = K_{D\tau min}||s_\tau|| + k_\tau \left( ||\widetilde{\boldsymbol{p}}_\tau|| - \frac{p_{\tau\,max}}{2} \right)^2 - k_\tau \frac{p_{\tau\,max}^2}{4}, \tag{53}$$

$$P_{eii} K_{Dnii} |s_{ni}| + P_{eii} k_{ni} ||\widetilde{\boldsymbol{p}}_{ni}|| (||\widetilde{\boldsymbol{p}}_{ni}|| - p_{ni\,max}) = P_{eii} K_{Dnii} |s_{ni}| + P_{eii} k_{ni} \left( ||\widetilde{\boldsymbol{p}}_{ni}|| - \frac{p_{ni\,max}}{2} \right)^2 - P_{eii} k_{ni} \frac{p_{ni\,max}^2}{4}. \tag{54}$$

They are positive if the errors meet the following inequalities:

$$||s_\tau|| > \frac{k_\tau p_{\tau\,max}^2}{4 K_{D\tau min}} \equiv b_{s\tau}, \tag{55}$$

$$|s_{ni}| > \frac{k_{ni} p_{ni\,max}^2}{4 K_{Dnii}} \equiv b_{sni}, \tag{56}$$

or:

$$||\widetilde{\boldsymbol{p}}_\tau|| > p_{\tau\,max} \equiv b_{p\tau}, \tag{57}$$

$$||\widetilde{\boldsymbol{p}}_{ni}|| > p_{ni\,max} \equiv b_{pni}, \tag{58}$$

It follows that $\dot{V}$ is negative outside the compact sets defined by Equations (55)–(58). According to the extension of the standard Lyapunov theory [36–38], it can be concluded that $||s_\tau||$, $|s_{ni}|$, $||\widetilde{\boldsymbol{p}}_\tau||$, and $||\widetilde{\boldsymbol{p}}_{ni}||$ are uniformly ultimately bounded and the control system is stable. Therefore, the filtered tracking error $s$ and its derivative are limited, similar to the vectors of parameter estimates $\hat{\boldsymbol{p}}_\tau$ and $\hat{\boldsymbol{p}}_{ni}$.  □

**Remark 1.** *Equations (34) and (35) actually depend on the signum function as required by strict mathematical proof. To avoid chattering, the steeply sloped hyperbolic tangent function can be used in simulations and practical applications to approximate the signum function.*

**Remark 2.** *The algorithm does not require assumptions about the inaccuracy of the actual surface.*

## 5. Simulation Results

A model of a robotic manipulator was constructed, whose kinematic structure is shown in Figure 5. It is a robot with a three-link arm, at the end of which there is an end-effector with two degrees of freedom. The arm is used to reach the desired position, and the orientation of the robot's tool is accomplished by adjusting the end-effector. The end-effector is not driven directly by motors mounted on the robot arm, but by motors located in the base using a drive train. This results in the fact that the orientation of the end-effector does not depend on the position of the arm. Thus, the orientation of the end-effector can be set by locking its drives in the selected positions. In this situation, the

robotic manipulator can be treated as a system with three degrees of freedom. Details on kinematics, dynamics, and parameters used in the simulation are given in Appendix A.

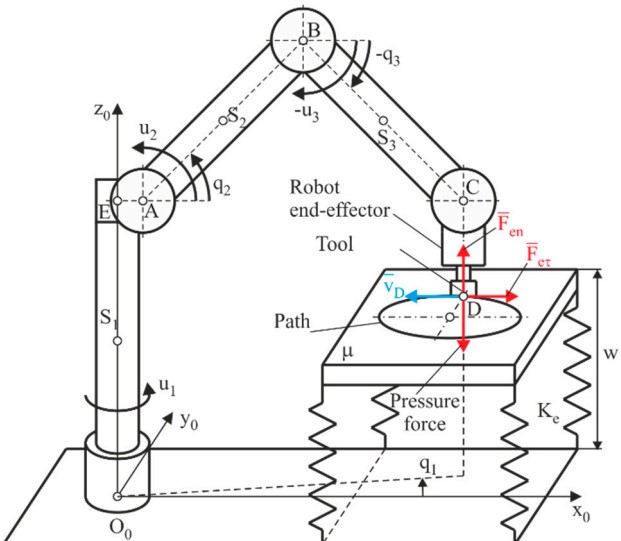

**Figure 5.** Model of a robotic manipulator in contact with a flexible environment: $O_0E = d_1$, $EA = l_1$, $AB = l_2$, $Bc = l_3$, $CD = d_5$—geometrical parameters characterizing the robot arm; $q_1$, $q_2$, $q_3$—angles of link rotation assumed as generalized coordinates; $u_1$, $u_2$, $u_3$—input moments; $K_e$—stiffness coefficient, $\mu$—coefficient of dry friction.

It was assumed that the end-effector should move on a flat surface lying in the plane $\pi$ parallel to the plane $x_0y_0$ and distant from it by $w$. The end-effector of the robot should simultaneously exert pressure perpendicular to the contact surface.

The desired motion path of point D is shown in Figure 6a,b, and the profile of the desired velocity of motion is shown in Figure 6c. The positional trajectory obtained from the solution of the system of Equation (A26) at the assumed velocity of Equation (A27) is shown in Figure 7 along with the desired force trajectory in the normal direction to the contact surface. The presented control algorithm requires the nominal trajectory of motion in the normal direction $c_{n\ nom}$ to be given. This results from the assumed shape of the interaction surface and, in the case under consideration, has the form $c_{n\ nom} = w = const.$, $\dot{c}_{n\ nom} = 0$, $\ddot{c}_{n\ nom} = 0$. In fact, due to the pressure of the end-effector of the robot on the flexible contact surface, the condition $\dot{c}_n = 0$ will not be strictly maintained and there will be surface deformation proportional to the pressure force. Therefore, the expected deformation of the surface with the coefficient of stiffness $K_e$ under the influence of the desired force $F_{end}(t)$ is additionally taken into account, which is expressed by the variable $\delta_0 = F_{end}/K_e$.

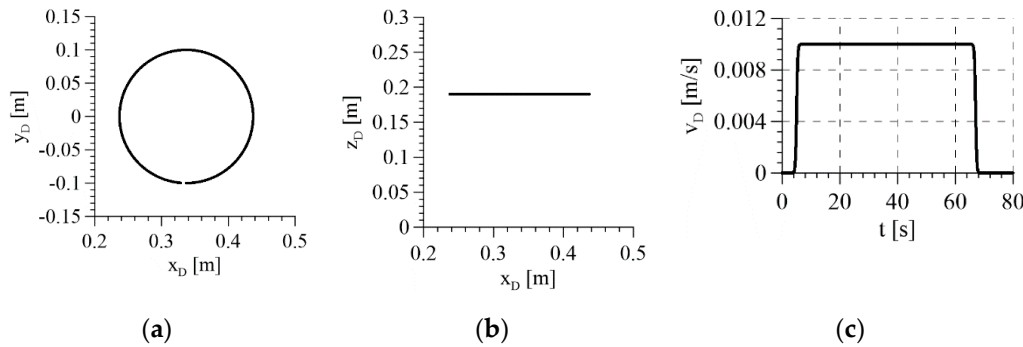

**Figure 6.** The desired motion: (**a**) motion path of point D in the xy plane; (**b**) motion path of point D in the xz plane; (**c**) the desired velocity of motion of point D.

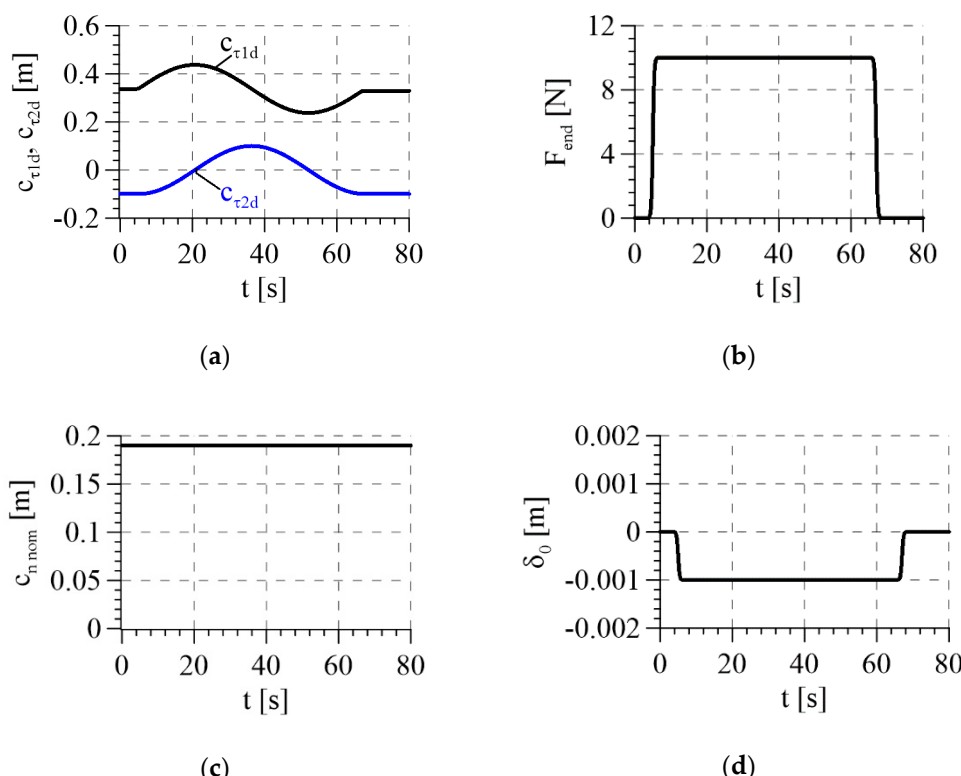

**Figure 7.** The desired trajectory: (**a**) coordinates of point D; (**b**) pressure force; (**c**) nominal coordinate of point D in tangential direction; (**d**) deformation of the surface under the influence of pressure force.

To test the properties of the system in the event of an inaccuracy in the surface of constraints, a simulation was carried out assuming a disturbance of constraints consisting in a depression in the surface of 0.001 m, which is shown in Figure 8a. Changing the surface of constraints can also be understood as a change in the surface profile as a function of time during the motion of the robot end-effector (Figure 8b).

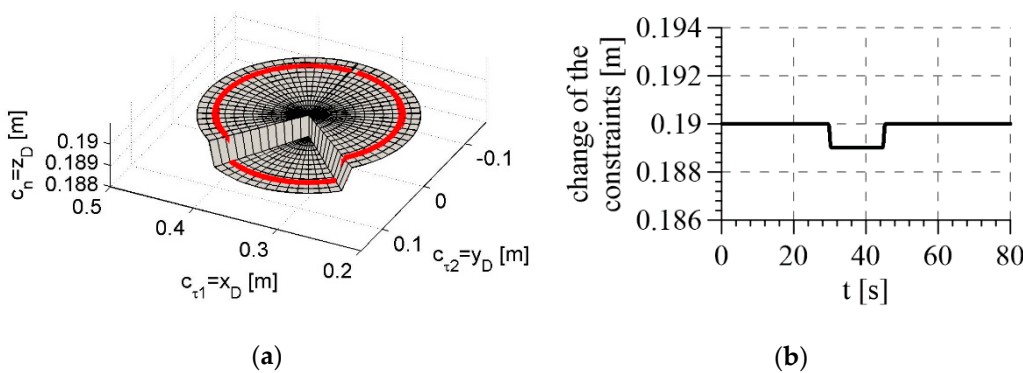

**Figure 8.** Disruption of the surface of constraints: (**a**) defect in the surface; (**b**) change of the surface of constraints in time along the desired motion path.

The overall control signal in the task space is shown in Figure 9. At the moment of surface disturbance, the control changes in such a way that the pressure of the end-effector on the surface is smaller (from 30 to 45 s). The signals generated by individual control subsystems are shown in Figure 10. According to the control law Equation (17), they are

in turn: PD control (Figure 10a,b), compensatory control (Figure 10c,d), interaction force compensation (Figure 10e,f), and robust control (Figure 10g,h).

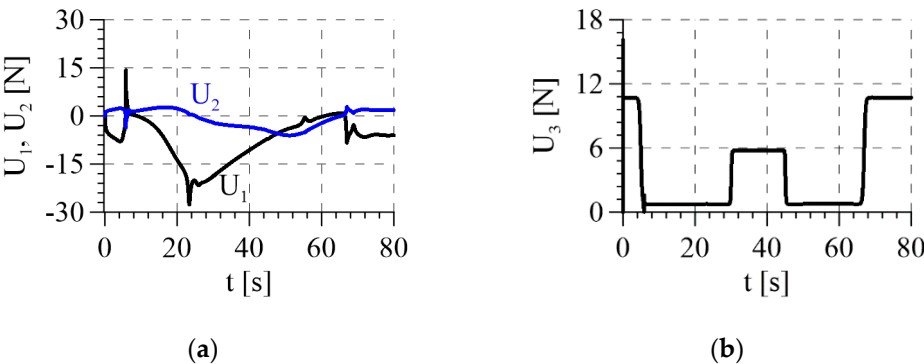

**Figure 9.** The overall control signals: (**a**) in tangential directions; (**b**) in the normal direction.

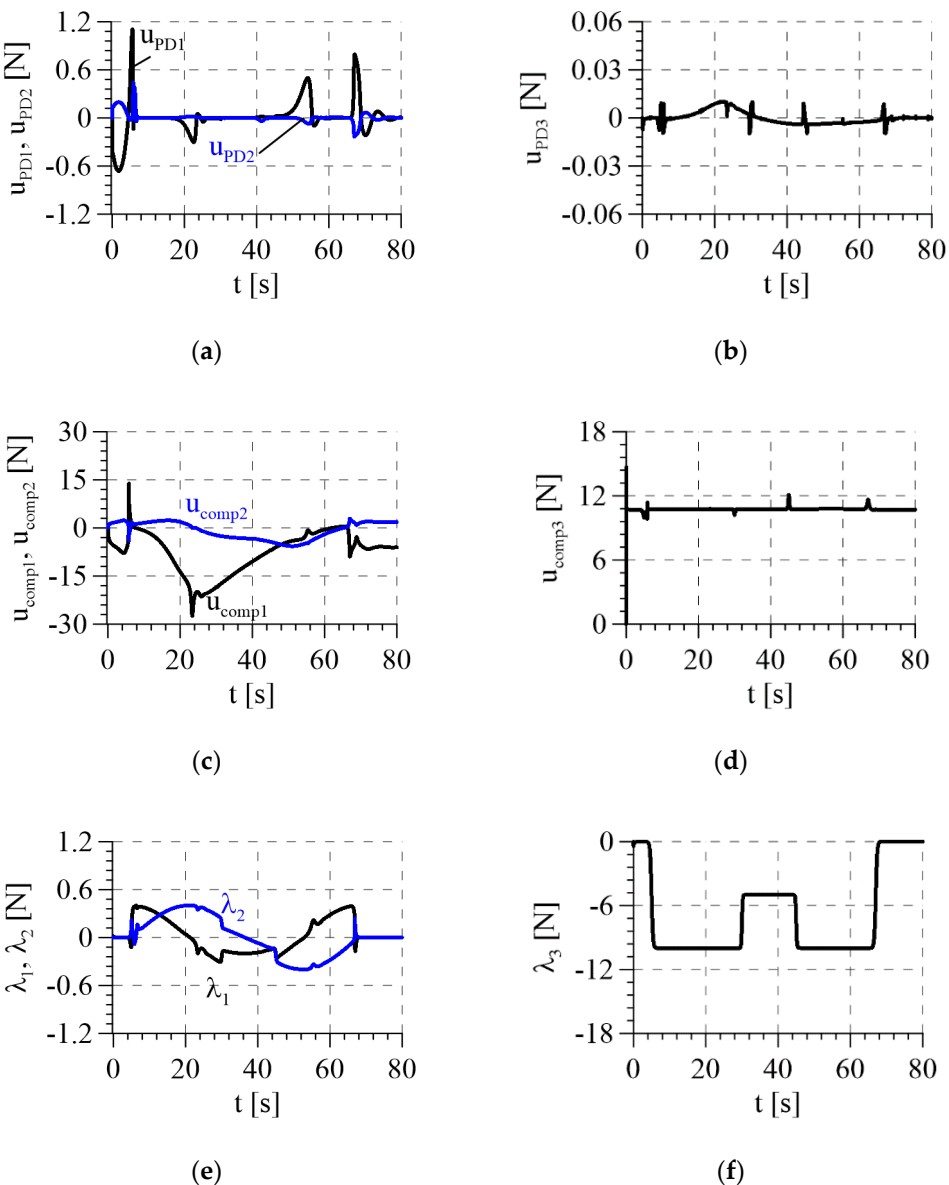

**Figure 10.** *Cont.*

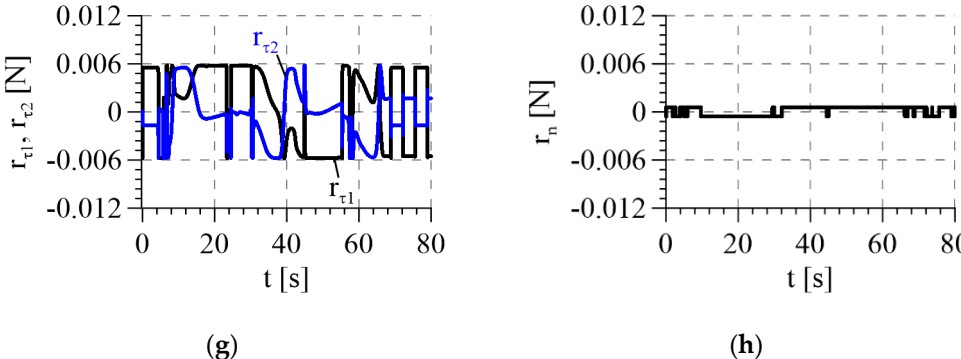

(g)    (h)

**Figure 10.** Control signals: (**a**) PD control in tangential directions, where $u_{PD1} = K_{D\tau1}s_{\tau1}$, $u_{PD2} = K_{D\tau2}s_{\tau2}$; (**b**) PD control in the normal direction, where $u_{PD3} = K_{Dn}s_n$; (**c**) compensatory control in tangential directions, where $u_{komp1} = \hat{f}_{\tau1}$, $u_{komp2} = \hat{f}_{\tau2}$; (**d**) compensatory control in the normal direction, where $u_{komp3} = \hat{f}_n$; (**e**) control compensating for the influence of friction forces; (**f**) control compensating for normal force; (**g**) robust control in tangential directions; (**h**) robust control in the normal direction.

The largest control changes in the area of surface disturbance concern PD control (Figure 10b), which contains a signal dependent on the surface inaccuracies and the control compensating for the effect of normal force $\Lambda_3 = F_{en}$ (Figure 10f), affecting the total control changes described earlier. The remaining control signals then show, at most, only slight oscillations.

Figure 11 presents the realized positional trajectory in the tangential directions (Figure 11a), the force trajectory in the normal direction (Figure 11c), and the deformation of the surface resulting from the implementation of this force (Figure 11b). In addition, Figure 11b shows the profile of the surface without deformation, and Figure 11c shows the desired pressure force (marked with dashed lines). Figure 11d shows deviation of the robot's end-effector from the assumed constraints in the normal direction.

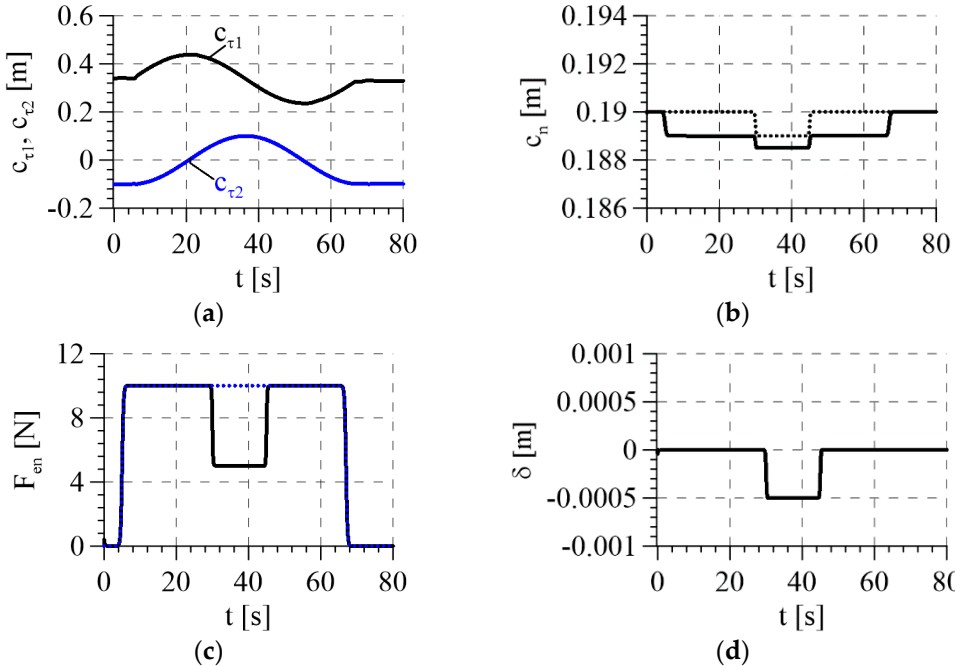

(a)    (b)

(c)    (d)

**Figure 11.** Realized trajectory: (**a**) coordinates of point D in the tangential directions; (**b**) coordinates of point D in the normal direction related to surface deformation; (**c**) pressure force; (**d**) deviation of robot's end-effector from assumed constraints in the normal direction.

As a result of the pressure force causing deformation in the normal direction, the actual motion path of the robot's end-effector in the phases of realizing the down force is shifted in relation to the assumed path. It should also be noted that in the area of surface disturbance, the end-effector displacement in the normal direction is not proportional to the desired force. The actual pressure force was reduced and, as a result, the increase in the end-effector displacement was less than that of the surface disruption. In real applications, this system action will prevent the deepening of surface losses.

The control errors obtained are presented in Figure 12. Regarding the tracking error for the force trajectory in the normal direction, it should be stated that at the moment of surface disturbance, the value of force error increases (Figure 12b) and simultaneously the deviation of the manipulator end-effector increases from the assumed constraints (Figure 11d). Therefore, the goal of minimizing the force error or the purpose of following the end-effector along a desired path in the normal direction are not fully realized. These goals are competitive and cannot be implemented simultaneously in a situation of surface disruption. However, it should be noted that the definition of a filtered tracking error in Equation (11) indicates that the simultaneous occurrence of force deviation and position deviation in the normal direction does not contradict the possibility of minimizing this error (Figure 12d). This approach favors finding a "balance" between the demands to minimize the force error and minimize the deviation from the nominal surface of the constraints.

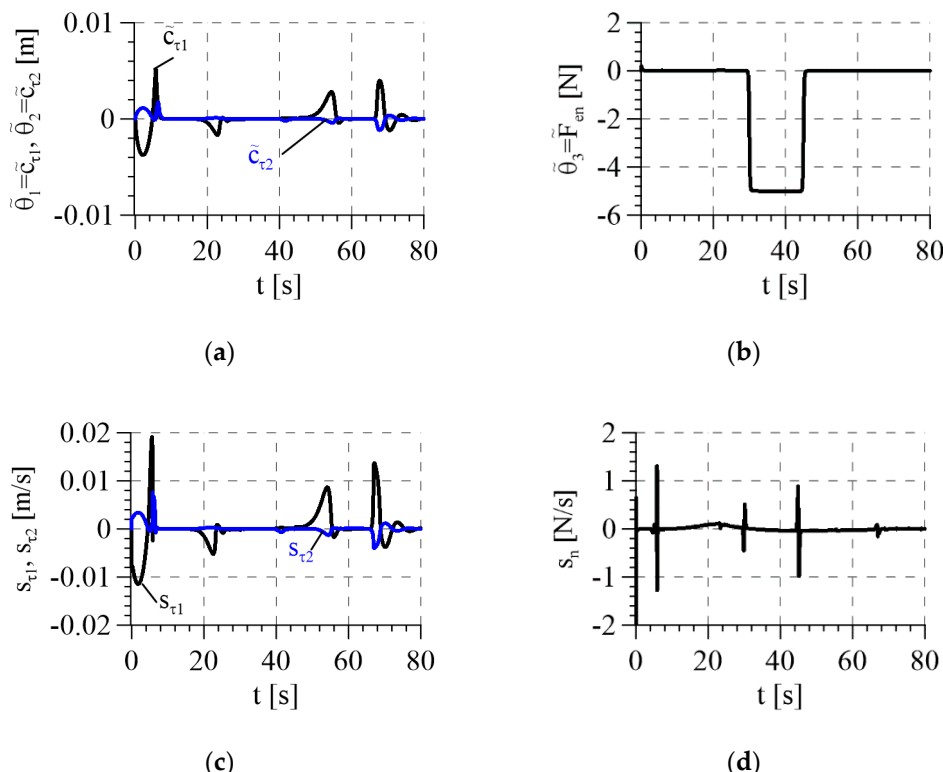

**Figure 12.** Tracking errors: (**a**) motion errors in tangential directions; (**b**) normal force error; (**c**) filtered motion errors in tangential directions; (**d**) filtered force error in the normal direction.

Estimates of robotic manipulator parameters are shown in Figure 13. The analysis of their waveforms shows that they are limited, in accordance with the proof of stability. The biggest changes in the parameter estimates occur in the initial phase of the motion when the control errors are greatest, and then the estimates are stabilized.

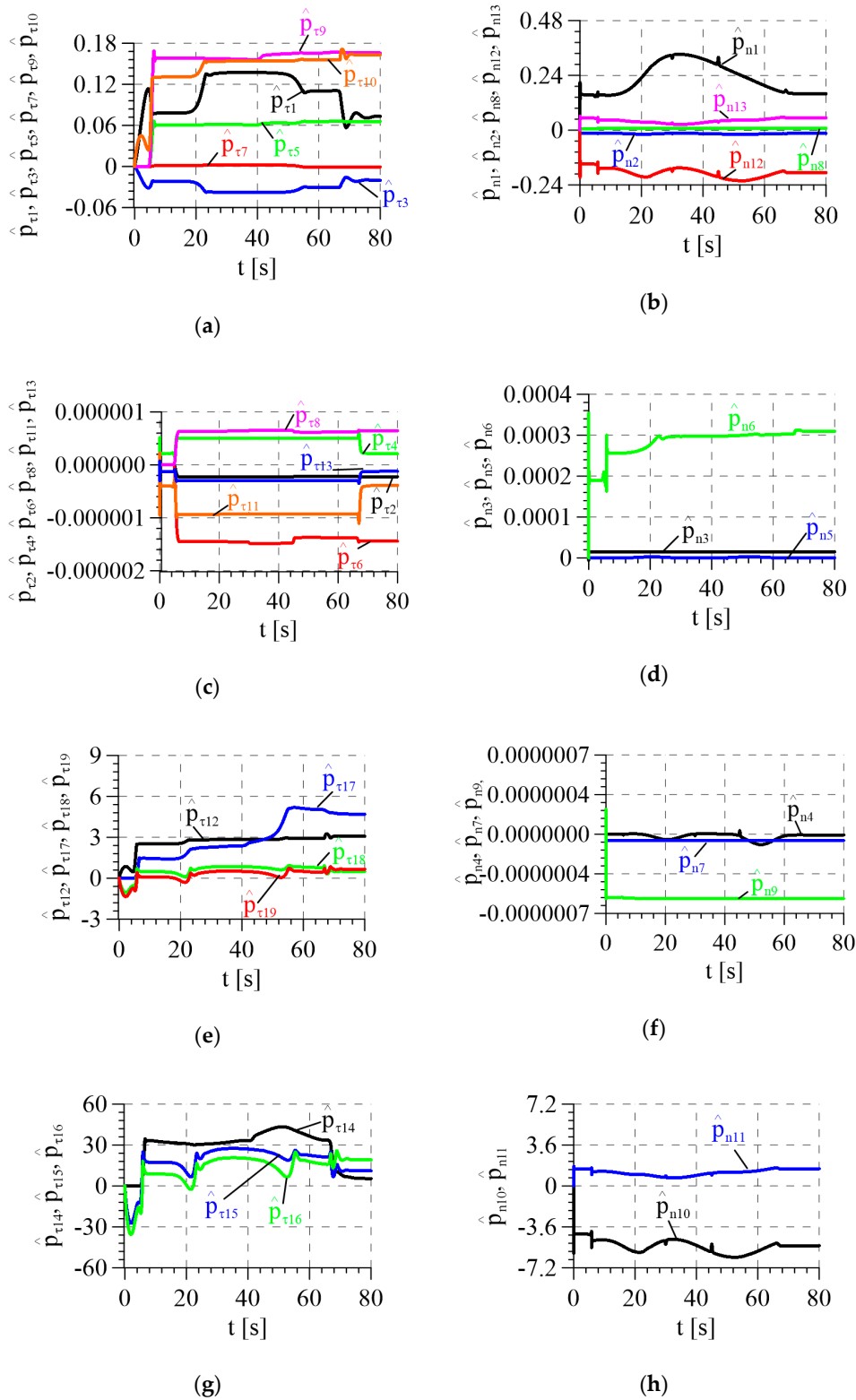

**Figure 13.** Estimates of the system parameters: (**a**) parameters $\hat{p}_{\tau 1}$, $\hat{p}_{\tau 3}$, $\hat{p}_{\tau 5}$, $\hat{p}_{\tau 7}$, $\hat{p}_{\tau 9}$, $\hat{p}_{\tau 10}$; (**b**) parameters $\hat{p}_{n1}$, $\hat{p}_{n2}$, $\hat{p}_{n8}$, $\hat{p}_{n12}$, $\hat{p}_{n13}$ (**c**) parameters $\hat{p}_{\tau 2}$, $\hat{p}_{\tau 4}$, $\hat{p}_{\tau 6}$, $\hat{p}_{\tau 8}$, $\hat{p}_{\tau 11}$, $\hat{p}_{\tau 13}$; (**d**) parameters $\hat{p}_{n3}$, $\hat{p}_{n5}$, $\hat{p}_{n6}$; (**e**) parameters $\hat{p}_{\tau 12}$, $\hat{p}_{\tau 17}$, $\hat{p}_{\tau 18}$, $\hat{p}_{\tau 19}$; (**f**) parameters $\hat{p}_{n4}$, $\hat{p}_{n7}$, $\hat{p}_{n9}$; (**g**) parameters $\hat{p}_{\tau 14}$, $\hat{p}_{\tau 15}$, $\hat{p}_{\tau 16}$; (**h**) parameters $\hat{p}_{n10}$, $\hat{p}_{n11}$.

This section presents the results of simulation tests with a description, with particular emphasis on the impact of the selected constraint distortion model on the control of a robotic manipulator. The test results indicate that the requirements for the control system

were met. The control system is stable and knowledge of the stiffness coefficient of the environment is not required. In addition, the control system operates in the prescribed manner in the presence of a surface disturbance, providing a compromise between the implementation of the desired force and following the desired path. Depending on the chosen value of the co-operation coefficient $w_\delta$, priority is given to achieve the desired force or nominal motion path. This is shown in Figure 14, which results from the additional simulations performed.

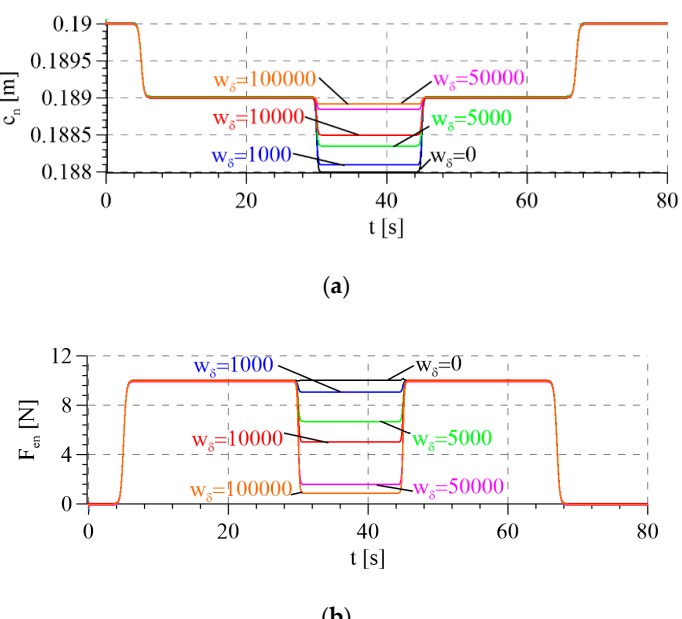

**Figure 14.** The influence of the co-operation gain coefficient $w_\delta$ on the task of the robotic manipulator: (**a**) the coordinates of point D in the normal direction; (**b**) the pressure force.

Figure 14 shows the influence of the $w_\delta$ gain on the operation of the control system. The graphs presented in this figure were obtained as a result of the simulation of the adaptive system operation at different values of the coefficient $w_\delta$. Its zero value results in the implementation of a desired force regardless of surface disturbances, i.e., classic force control is implemented. The control algorithm is then identical to that shown in [19,33]. In turn, an increase in the value of coefficient $w_\delta$ causes an increase in the importance of maintaining the nominal trajectory of motion in the normal direction despite surface disturbances, which takes place at the expense of the accuracy of force realization. It should be noted that the effects of the system's action are not proportional to the increase in the value of coefficient $w_\delta$ and it is not possible to switch the system to the strategy of controlling only the position, because this would require an infinitely large value of $w_\delta$.

## 6. Conclusions

This article presents an approach to positional force control, taking into account the inaccuracy of constraints, the implementation of which requires the adoption of a desired trajectory of motion and the course of the desired pressure force, in addition to the assumptions of the nominal shape of the surface of the environment. These assumptions introduce a difference between the method presented in the article and the standard control strategies used in position/force control. This approach makes it possible to implement force control in the normal direction to the surface of the environment, which is characterized by a kind of "flexibility" in the implementation of two elementary strategies. Introducing reactions of virtual constraints provides automatic adjustment of the robot interaction force with a susceptible environment, minimizing the impact of geometric inaccuracy of the environment.

The novelty of the presented solution is based on introducing an additional module to the control law in directions normal to the interaction surface, which allows for a fluent change of control strategy in the case of occurrence of distortions in the surface. By introducing one additional cooperation amplification factor $w_\delta$ to the classical strategy for force control, the strategy for maintaining nominal position was added, reducing the importance of the force error minimization in the case of distortions in the shape of the surface. This means of reaction of the control system is vital in the case of the interaction of a robot with inaccurately described surfaces.

An additional module in the control law may be perceived as a virtual viscotic resistance force and resilient environment acting upon the robot. Such an interpretation facilitates the intuitive selection of amplifications and allows for foreseeing the behavior of the system in case disturbances occur. Introducing the factor that amplifies the strategy for maintaining the nominal trajectory by the activity of virtual constraints reaction force allows for automatic adjustment of the interaction force of the robot with a susceptible environment.

The proposed algorithm contains a non-linear function (Equation (25)) dependent on variable $v$ and its derivative $\dot{v}$, which in turn is a function of, inter alia, $\dot{\delta}$ and $\ddot{\delta}$. The velocity error of the end-effector $\dot{\delta}$ can be determined based on the velocity measurements of the arm links and kinematics equations. To determine the variable $\ddot{\delta}$, which depends on the actual acceleration of the robot end-effector $\ddot{c}_n$, a filtered and differential velocity signal $\dot{c}_n$ can be used in the case of a slightly noisy signal, or the signal from an additional acceleration sensor in the robot's end-effector can be used.

**Funding:** This research received no external funding.

**Institutional Review Board Statement:** Not applicable.

**Informed Consent Statement:** Not applicable.

**Data Availability Statement:** Data sharing not applicable.

**Conflicts of Interest:** The author declares no conflict of interest.

**Appendix A**

The dynamics of a robotic manipulator in interactions with a flexible environment is described by Equation (1), where matrices and vectors have the following form [19]:

$$\left.\begin{aligned}
A(q) &= \left(JM(q)^{-1}J^T\right)^{-1}\\
H(q,\dot{q}) &= J^{-T}C(q,\dot{q})J^{-1} - A(q)\dot{J}J^{-1}\\
B(q,\dot{q}) &= J^{-T}\left(F(\dot{q}) + G(q)\right)\\
\Psi(q,t) &= J^{-T}\xi(t)\\
U &= J^{-T}u
\end{aligned}\right\} , \tag{A1}$$

Kinematics equations defining the position of the robot's end-effector in the base coordinate system have the following form:

$$c = k(q) = \begin{bmatrix} x_D \\ y_D \\ z_D \end{bmatrix} = \begin{bmatrix} (l_1 + l_2 cosq_2 + l_3 cosq_3)cosq_1 \\ (l_1 + l_2 cosq_2 + l_3 cosq_3)sinq_1 \\ d_1 + l_2 sinq_2 + l_3 sinq_3 - d_5 \end{bmatrix} , \tag{A2}$$

where: $x_D$, $y_D$, $z_D$—coordinates of point D in the reference system; $l_1$, $l_2$, $l_3$, $d_1$, $d_5$—geometric parameters of the arm; $q_1$, $q_2$, $q_3$—robot configuration coordinates (angles of rotation of the links).

Simplifying the description of kinematics to the three-dimensional task space $xyz$ without taking into account the orientation of the end-effector, one can consider the system with the three degrees of freedom resulting from the mobility of the arm, ignoring the

mobility of the end-effector. The analytical Jacobian $J$ determined using Equations (A2) has the following form:

$$J = \begin{bmatrix} -(l_1 + l_2 cosq_2 + l_3 cosq_3)sinq_1 & -l_2 sinq_2 cosq_1 & -l_3 sinq_3 cosq_1 \\ (l_1 + l_2 cosq_2 + l_3 cosq_3)cosq_1 & -l_2 sinq_2 sinq_1 & -l_3 sinq_3 sinq_1 \\ 0 & l_2 cosq_2 & l_3 cosq_3 \end{bmatrix}. \quad \text{(A3)}$$

Matrices and vectors $M(q), C(q,\dot{q}), F(\dot{q}), G(q)$ results from the description of robot dynamics in joint space [19], and in the case of the analysed three-link robot are given by Equations (A4)–(A12):

$$M(q) = \begin{bmatrix} M_{11} & 0 & 0 \\ 0 & p_6 & p_2 l_2 cos(q_3 - q_2) \\ 0 & p_2 l_2 cos(q_3 - q_2) & p_7 \end{bmatrix}. \quad \text{(A4)}$$

$$M_{11} = 2p_1 l_1 cosq_2 + 2p_2(l_1 + l_2 cosq_2)cosq_3 + 0.5p_3 cos(2q_2) + 0.5p_4 cos(2q_3) + p_5, \quad \text{(A5)}$$

$$C(q,\dot{q}) = \begin{bmatrix} -b\dot{q}_2 - c\dot{q}_3 & -b\dot{q}_1 & -c\dot{q}_1 \\ b\dot{q}_1 & 0 & -p_2 l_2 sin(q_3 - q_2)\dot{q}_3 \\ c\dot{q}_1 & p_2 l_2 sin(q_3 - q_2)\dot{q}_2 & 0 \end{bmatrix}, \quad \text{(A6)}$$

$$b = p_1 l_1 sinq_2 + p_2 l_2 sinq_2 cosq_3 + 0.5p_3 sin(2q_2), \quad \text{(A7)}$$

$$c = p_2(l_1 + l_2 cosq_2)sinq_3 + 0.5p_4 sin(2q_3), \quad \text{(A8)}$$

$$F(\dot{q}) = \begin{bmatrix} p_8 \dot{q}_1 + p_{11} sign(\dot{q}_1) \\ p_9 \dot{q}_2 + p_{12} sign(\dot{q}_2) \\ p_{10} \dot{q}_3 + p_{13} sign(\dot{q}_3) \end{bmatrix}, \quad \text{(A9)}$$

$$G(q) = \begin{bmatrix} 0 \\ p_1 g cosq_2 \\ p_2 g cosq_3 \end{bmatrix}, \quad \text{(A10)}$$

$$\xi(t) = \begin{bmatrix} \xi_1 \\ \xi_2 \\ \xi_3 \end{bmatrix}, \quad \text{(A11)}$$

$$u = \begin{bmatrix} u_1 \\ u_2 \\ u_3 \end{bmatrix}. \quad \text{(A12)}$$

Parameters in matrices and vectors describing the dynamic properties of the system are as follows:

$$\left. \begin{aligned} p_1 &= l_{c2}m_2 + l_2 m_3 + l_2 m_D \\ p_2 &= l_{c3}m_3 + l_3 m_D \\ p_3 &= l_{c2}^2 m_2 + l_2^2 m_3 + l_2^2 m_D - I_{2xx} + I_{2yy} \\ p_4 &= l_{c3}^2 m_3 + l_3^2 m_D - I_{3xx} + I_{3yy} \\ p_5 &= 0.5(I_{2xx} + I_{2yy} + I_{3xx} + I_{3yy}) + I_{1yy} + (l_1^2 + 0.5l_{c2}^2)m_2 + \\ &\quad + (l_1^2 + 0.5l_2^2 + 0.5l_{c3}^2)m_3 + (l_1^2 + 0.5l_2^2 + 0.5l_3^2)m_D \\ p_6 &= l_{c2}^2 m_2 + l_2^2 m_2 + l_2^2 m_D + I_{2zz} \\ p_7 &= l_{c3}^2 m_3 + l_3^2 m_D + I_{3zz} \\ p_8 &= F_{v1} \\ p_9 &= F_{v2} \\ p_{10} &= F_{v3} \\ p_{11} &= F_{c1} \\ p_{12} &= F_{c2} \\ p_{13} &= F_{c3} \end{aligned} \right\} \quad \text{(A13)}$$

where: $m_i$—mass of the $i$th link, $m_D$—mass of the end-effector, $l_i$—length of the $i$th link, $l_{ci}$—distance between the center of mass of the $i$th link and the end of the $i-1$ link, $I_{i(...)}$—mass moment of inertia of the $i$th link relative to the appropriate axis, $F_{vi}$—viscous friction coefficient in the $i$th kinematic pair, $F_{ci}$—moment of dry friction forces in the $i$th kinematic pair.

The task space of the robotic manipulator $\{C\}$ was separated into the tangential subspace $\{T\}$ and normal subspace $\{N\}$, taking two tangential directions, $c_{\tau 1} = x_D$ and $c_{\tau 2} = y_D$, and one normal direction, $c_n = z_D$, which were included in the kinematics equation:

$$
c = \begin{bmatrix} x_D \\ y_D \\ z_D \end{bmatrix} = \begin{bmatrix} c_{\tau 1} \\ c_{\tau 2} \\ c_n \end{bmatrix} = \begin{bmatrix} (l_1 + l_2 cosq_2 + l_3 cosq_3)cosq_1 \\ (l_1 + l_2 cosq_2 + l_3 cosq_3)sinq_1 \\ d_1 + l_2 sinq_2 + l_3 sinq_3 - d_5 \end{bmatrix}. \tag{A14}
$$

The velocity of the end-effector in the case under consideration is a three-dimensional vector:

$$
\dot{c} = \begin{bmatrix} \dot{c}_{\tau 1} \\ \dot{c}_{\tau 2} \\ \dot{c}_n \end{bmatrix}, \tag{A15}
$$

and its value is determined by the formula:

$$
\dot{c} = \sqrt{\dot{c}_{\tau 1}^2 + \dot{c}_{\tau 2}^2 + \dot{c}_n^2}. \tag{A16}
$$

It was assumed that the interaction forces of the robot with the environment in task coordinates are described by the following equations:

$$
\Lambda = \begin{bmatrix} F_{e\tau 1} \\ F_{e\tau 2} \\ F_{en} \end{bmatrix} = \begin{bmatrix} \mu F_{en} \dfrac{-\dot{c}_{\tau 1}}{\sqrt{\dot{c}_{\tau 1}^2 + \dot{c}_{\tau 2}^2}} \\ \mu F_{en} \dfrac{-\dot{c}_{\tau 2}}{\sqrt{\dot{c}_{\tau 1}^2 + \dot{c}_{\tau 2}^2}} \\ K_e c_n \end{bmatrix}. \tag{A17}
$$

in which $\mu F_{en}$ is the value of the friction force, $K_e c_n$ is the pressure force, $\dfrac{\dot{c}_{\tau 1}}{\sqrt{\dot{c}_{\tau 1}^2 + \dot{c}_{\tau 2}^2}}$ and $\dfrac{\dot{c}_{\tau 2}}{\sqrt{\dot{c}_{\tau 1}^2 + \dot{c}_{\tau 2}^2}}$ are the cosine and sine of the angle between the friction force and the $\tau_1$ axis, and the "−" signs in the counters of the first two elements result from the opposite return of the friction force relative to the velocity. The first element of the interaction force vector is therefore a projection of the friction force vector on the $\tau_1$ axis, and the second element of this vector is a projection on the $\tau_2$ axis. The term $\mu F_{en} \dfrac{-\dot{c}_{\tau 1}}{\sqrt{\dot{c}_{\tau 1}^2 + \dot{c}_{\tau 2}^2}}$ is equivalent to the expression $\mu F_{en} \dfrac{-|\dot{c}_{\tau 1}|sign(\dot{c}_{\tau 1})}{\sqrt{\dot{c}_{\tau 1}^2 + \dot{c}_{\tau 2}^2}}$, similarly $\mu F_{en} \dfrac{-\dot{c}_{\tau 2}}{\sqrt{\dot{c}_{\tau 1}^2 + \dot{c}_{\tau 2}^2}}$ is equivalent to $\mu F_{en} \dfrac{-|\dot{c}_{\tau 2}|sign(\dot{c}_{\tau 2})}{\sqrt{\dot{c}_{\tau 1}^2 + \dot{c}_{\tau 2}^2}}$.

Parameters occurring in the dynamic equations of motion of the robotic manipulator–environment system are given in Table A1.

**Table A1.** Parameters of the robot and the environment used in simulation tests.

| Parameter | Unit | Value |
|---|---|---|
| $p_1$ | kg·m | 0.390 |
| $p_2$ | kg·m | 0.108 |
| $p_3$ | kg·m$^2$ | 0.678 |
| $p_4$ | kg·m$^2$ | 0.384 |
| $p_5$ | kg·m$^2$ | 0.684 |
| $p_6$ | kg·m$^2$ | 0.678 |

**Table A1.** *Cont.*

| Parameter | Unit | Value |
|:---:|:---:|:---:|
| $p_7$ | kg·m$^2$ | 0.390 |
| $p_8$ | N·ms | 31.65 |
| $p_9$ | N·ms | 31.39 |
| $p_{10}$ | N·ms | 31.41 |
| $p_{11}$ | N·m | 1.170 |
| $p_{12}$ | N·m | 1.092 |
| $p_{13}$ | N·m | 1.098 |
| $d_1$ | m | 0.35 |
| $l_1$ | m | 0.026 |
| $l_2$ | m | 0.22 |
| $l_3$ | m | 0.22 |
| $d_5$ | m | 0.16 |
| $K_e$ | N/m | 10,000 |
| $\mu$ | - | 0.04 |

The force $F_{en}$ is one of the controlled variables, therefore it is required to adopt a desired force trajectory in the normal direction $F_{end}(t) \in R^1$, $\dot{F}_{end}(t)$, $\ddot{F}_{end}(t)$. The desired force depends on the process, the implementation of which requires an appropriate pressure force. Usually, in machining processes, there are assumed to be intervals of constant pressure force, possibly with a transitional period in which the force is to be smoothly increased or reduced to an appropriate value. The tests assume a pressure force with a defined maximum value $F_{end\ max}$, which is to be smoothly achieved, and is ensured by assuming the desired pressure force according to the formula:

$$F_{end} = \frac{F_{end\ max}}{1 + exp[-w_n(t - t_{ns})]} - \frac{F_{end\ max}}{1 + exp[-w_n(t - t_{nk})]}, \tag{A18}$$

where $F_{end\ max}$ is the maximum pressure force, $w_n > 0$ is the coefficient related to the rate of increase and decrease in force, $t_{ns}$ and $t_{nk}$ determine the time of increase and decrease in force, $t \in (0, 70)$ s. The desired force meets the limit $|F_{end}| \leq F_{end\ max}$, and its first and second derivatives with respect to time are limited so that $|\dot{F}_{end}| \leq F_{end\ max}w_n$ and $|\ddot{F}_{end}| \leq F_{end\ max}w_n^2$. Equation (A18) ensures that in the normal direction the desired pressure force is continuously non-negative and has a continuous first and second derivative with respect to time.

The next controlled variables are the position and velocity of the end-effector in the contact plane. It is known that the possible velocity must be tangential to the motion path, i.e., to satisfy the equation:

$$grad(h(c))\dot{c} = 0, \tag{A19}$$

where $h(c) = \mathbf{0}$ is the equation of the motion path. To describe a path in a three-dimensional space requires two equations, i.e., $h_1(c) = 0$ and $h_2(c) = 0$, so it can be written as: $h(c) = \begin{bmatrix} h_1(c) & h_2(c) \end{bmatrix}^T = \mathbf{0}$. In the analyzed example, the contact surface is two-dimensional, so there are two tangential directions along which the end-effector of the robot can move. It was assumed that the motion takes place in a circle with the given equation:

$$h_1(c) = (c_{\tau 1} - x_O)^2 + c_{\tau 2}^2 - R^2 = 0, \tag{A20}$$

where $x_O$ determines the position of the circle centre and $R$ is the circle radius. The circle lies in the $\pi$ plane given by the equation:

$$h_2(c) = c_n - w = \mathbf{0}, \tag{A21}$$

where $w$ determines the distance of the contact plane from the plane $x_O y_O$ (Figure 5). The condition (A19) can be written as follows:

$$
\begin{bmatrix} \frac{\partial h_1(c)}{\partial c_{\tau 1}} & \frac{\partial h_1(c)}{\partial c_{\tau 2}} & \frac{\partial h_1(c)}{\partial c_n} \\ \frac{\partial h_2(c)}{\partial c_{\tau 1}} & \frac{\partial h_2(c)}{\partial c_{\tau 1}} & \frac{\partial h_2(c)}{\partial c_n} \end{bmatrix} \begin{bmatrix} \dot{c}_{\tau 1} \\ \dot{c}_{\tau 2} \\ \dot{c}_n \end{bmatrix} = 0.
\tag{A22}
$$

After taking Equations (A20) and (A21) into account, Equation (A22) was written in the form:

$$
\begin{bmatrix} 2(c_{\tau 1} - x_O) & 2c_{\tau 2} & 0 \\ 0 & 0 & 1 \end{bmatrix} \begin{bmatrix} \dot{c}_{\tau 1} \\ \dot{c}_{\tau 2} \\ \dot{c}_n \end{bmatrix} = 0,
\tag{A23}
$$

and then as a system of two equations:

$$
\begin{cases} 2(c_{\tau 1} - x_O)\dot{c}_{\tau 1} + 2c_{\tau 2}\dot{c}_{\tau 2} = 0 \\ \dot{c}_n = 0 \end{cases}.
\tag{A24}
$$

The second equation of the System (A24) results in the velocity in the normal direction $\dot{c}_n = 0$, which means that there is no planned motion of the robot end-effector in this direction, and the velocity components in the tangential directions must fulfil the first equations of the System (A24). In addition, they must satisfy Equation (A16), taking into account that $\dot{c}_n = 0$. This gives another set of equations:

$$
\begin{cases} 2(c_{\tau 1} - x_O)\dot{c}_{\tau 1} + 2c_{\tau 2}\dot{c}_{\tau 2} = 0 \\ \dot{c} = \sqrt{\dot{c}_{\tau 1}^2 + \dot{c}_{\tau 2}^2} \end{cases},
\tag{A25}
$$

whose solution in the form:

$$
\begin{cases} \dot{c}_{\tau 1} = \pm \frac{c_{\tau 2}\dot{c}}{\sqrt{c_{\tau 2}^2 + (c_{\tau 1} - x_O)^2}} \\ \dot{c}_{\tau 2} = \mp \frac{(c_{\tau 1} - x_O)\dot{c}}{\sqrt{c_{\tau 2}^2 + (c_{\tau 1} - x_O)^2}} \end{cases},
\tag{A26}
$$

enables numerical calculation of velocity components $\dot{c}_{\tau 1}$ and $\dot{c}_{\tau 2}$, in addition to coordinates $c_{\tau 1}$ and $c_{\tau 2}$ with assumed initial conditions $c_{\tau 1}(0)$, $c_{\tau 2}(0)$, and velocity values $\dot{c}$ at any moment of time. By differentiating the Equation (A26), the accelerations in the tangential directions can be determined.

To calculate the desired trajectory of motion in the contact plane, i.e., $\mathbf{c}_{\tau d}(t) = \begin{bmatrix} c_{\tau d1} & c_{\tau d2} \end{bmatrix}^T$, $\dot{\mathbf{c}}_{\tau d}(t) = \begin{bmatrix} \dot{c}_{\tau d1} & \dot{c}_{\tau d2} \end{bmatrix}^T$, $\ddot{\mathbf{c}}_{\tau d}(t) = \begin{bmatrix} \ddot{c}_{\tau d1} & \ddot{c}_{\tau d2} \end{bmatrix}^T$, velocity $\dot{c}_d$ was assumed, whose value changes according to the following equation:

$$
\dot{c}_d = \frac{\dot{c}_{dmax}}{1 + exp[-w_\tau(t - t_{\tau s})]} - \frac{\dot{c}_{dmax}}{1 + exp[-w_\tau(t - t_{\tau k})]},
\tag{A27}
$$

where $\dot{c}_{dmax}$ is the maximum desired velocity, $w_\tau > 0$ is the coefficient related to the speed of acceleration and deceleration of the end-effector, $t_{\tau s}$ and $t_{\tau k}$ determine the acceleration and deceleration time of the end-effector, $t \in (0, 70)$ s. The desired velocity of motion meets the limit $|\dot{c}_d| \leq \dot{c}_{dmax}$, and the desired acceleration meets the condition $|\ddot{c}_d| \leq \dot{c}_{dmax} w_\tau$.

The parameters of the positional and force trajectory are listed in Table A2. The parameters of the control system are presented in Table A3.

**Table A2.** Parameters of the desired trajectory.

| Parameter | Unit | Value |
|:---:|:---:|:---:|
| $\dot{c}_{dmax}$ | m/s | 0.01 |
| $w_\tau$ | s$^{-1}$ | 5 |
| $t_{\tau s}$ | s | 5 |
| $t_{\tau k}$ | s | 67 |
| $F_{end\,max}$ | N | 10 |
| $w_n$ | s$^{-1}$ | 5 |
| $t_{ns}$ | s | 5 |
| $t_{nk}$ | s | 67 |
| $w$ | m | 0.19 |
| $R$ | m | 0.1 |
| $x_O$ | m | 0.3371 |
| $c_{\tau 1}(0)$ | m | 0.3371 |
| $c_{\tau 2}(0)$ | m | −0.1 |

**Table A3.** Parameters of the adaptive control system.

| Parameter | Unit | Value |
|:---:|:---:|:---:|
| $K_{D\tau 1}$ | kg/s | 1 |
| $K_{D\tau 2}$ | kg/s | 1 |
| $K_{Dn}$ | s | 0.002 |
| $\Lambda_{\tau 1}$ | s$^{-1}$ | 3 |
| $\Lambda_{\tau 2}$ | s$^{-1}$ | 3 |
| $\Lambda_n$ | s$^{-1}$ | 3.5 |
| $k_\tau$ | - | 0.01 |
| $k_n$ | - | 0.01 |
| $K_\tau$ | N | 0.0001 |
| $K_n$ | N | 0.00001 |
| $w_\delta$ | N/m | 10000 |

To implement the robot's task, a control given by Equation (58) was used in which the gain matrices have the form $\boldsymbol{K}_D = diag\{K_{D\tau 1}, K_{D\tau 2},\ K_{Dn}\}$, $\boldsymbol{\Lambda} = diag\{\Lambda_{\tau 1},\ \Lambda_{\tau 2}, \Lambda_n\}$, and in the analyzed case $w_\delta$ is a one-dimensional coefficient, which determines the behavior of the system in the presence of surface disturbances. The system's non-linearity approximation function is decomposed into components in the following way $\hat{\boldsymbol{f}} = \begin{bmatrix} \hat{\boldsymbol{f}}_\tau^T & \hat{f}_n \end{bmatrix}^T$, where $\hat{\boldsymbol{f}}_\tau = \boldsymbol{Y}_\tau(\boldsymbol{q}, \dot{\boldsymbol{q}}, \boldsymbol{v}, \dot{\boldsymbol{v}})\hat{\boldsymbol{p}}_\tau$ and $\hat{f}_n = \boldsymbol{Y}_n(\boldsymbol{q}, \dot{\boldsymbol{q}}, \boldsymbol{v}, \dot{\boldsymbol{v}})\hat{\boldsymbol{p}}_n$, in which there are estimates $\hat{\boldsymbol{p}}_\tau$ and $\hat{\boldsymbol{p}}_n$ of the vectors $\boldsymbol{p}_\tau$ and $\boldsymbol{p}_n$ defined by the formulae:

$$\boldsymbol{p}_\tau = \begin{bmatrix} p_1, p_1 P_e, p_2, p_2 P_e, p_3, p_3 P_e, p_4,\ p_4 P_e, p_5,\ p_6, p_6 P_e, p_7,\ p_7 P_e, p_8,\ p_9, p_{10}, p_{11}, p_{12}, p_{13} \end{bmatrix}^T, \tag{A28}$$

$$\boldsymbol{p}_n = [p_1, p_2,\ p_2 P_e, p_3,\ p_4,\ p_6,\ p_6 P_e, p_7,\ p_7 P_e, p_9,\ p_{10}, p_{12}, p_{13}]^T, \tag{A29}$$

where the coefficient of vulnerability of the environment $P_e$ is taken into account. Matrices of parameter adaptation gain $\boldsymbol{\Gamma}_\tau$ and $\boldsymbol{\Gamma}_n$ were selected as diagonal matrices:

$$\begin{aligned}\boldsymbol{\Gamma}_\tau = diag\{&6.5{\cdot}10^{-4};\ 6.5{\cdot}10^{-4};\ 1.8{\cdot}10^{-4};\ 1.8{\cdot}10^{-9};\ 11.3;\ 1.13{\cdot}10^{-4};\ 0.64;\ 6.4{\cdot}10^{-5};\ 11.4; \\ &0.113;\ 1.13{\cdot}10^{-9};\ 1.3;\ 6.5{\cdot}10^{-10};\ 5.3{\cdot}10^{3};\ 5.2{\cdot}10^{2};\ 5.2{\cdot}10^{2};\ 1.9{\cdot}10^{2};\ 18.2;\ 18.3\},\end{aligned} \tag{A30}$$

$$\begin{aligned}\boldsymbol{\Gamma}_n = diag\{&6.5{\cdot}10^{-5};\ 1.8{\cdot}10^{-4};\ 1.8{\cdot}10^{-8};\ 1.13; \\ &0.64;\ 0.113;\ 1.13{\cdot}10^{-9};\ 6.5{\cdot}10^{-2};\ 6.5{\cdot}10^{-8};\ 0.52;\ 0.52;\ 1.8{\cdot}10^{-2};\ 1.8{\cdot}10^{-2}\},\end{aligned} \tag{A31}$$

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
