# Peer review of "Adaptive Position/Force Control of a Robotic Manipulator in Contact with a Flexible and Uncertain Environment"

_robotics, doi:10.3390/robotics10010032_

Round 1

Reviewer 1 Report

The paper presents a novel force/position control method aiming to deal with susceptible environments, minimising the impact of geometric inaccuracy of the environments. The novelty of the presented solution is based on introducing an additional module to the control law in directions normal to the interaction surface, which allows for a fluent change of control strategy in the case of occurrence of distortions in the surface.

The paper is too verbose. There are some many textbook formulations and concepts. For example, Eqs. (2)-(4) are for the derivation of Jacobian, which are totally not necessary to be shown in the paper. Section 3 could be shortened to be one paragraph. The English language is also weird as there are some many long sentences that could be shortened as well.

Although there are no real robot experiments, the simulation and theories are well presented. I recommend to publish the paper after polishing.

Reviewer 2 Report

See enclosed file.

Round 2

Reviewer 2 Report

In general, appendices are presented at the end of the manuscript after the list of references.